# Accelerated Primal-Dual Gradient Method for Smooth and Convex-Concave Saddle-Point Problems with Bilinear Coupling

**Dmitry Kovalev**
KAUST[*]
dakovalev1@gmail.com

**Alexander Gasnikov**
MIPT,[†] ISP RAS,[‡] HSE[§]
gasnikov@yandex.ru

**Peter Richtárik**
KAUST
richtarik@gmail.com

## Abstract

In this paper we study the convex-concave saddle-point problem $\min_x \max_y f(x) + y^T \mathbf{A}x - g(y)$, where $f(x)$ and $g(y)$ are smooth and convex functions. We propose an Accelerated Primal-Dual Gradient Method (APDG) for solving this problem, achieving (i) an optimal linear convergence rate in the strongly-convex-strongly-concave regime, matching the lower complexity bound (Zhang et al., 2021), and (ii) an accelerated linear convergence rate in the case when only one of the functions $f(x)$ and $g(y)$ is strongly convex or even none of them are. Finally, we obtain a linearly convergent algorithm for the general smooth and convex-concave saddle point problem $\min_x \max_y F(x, y)$ without the requirement of strong convexity or strong concavity.

## 1 Introduction

In this paper we revisit the well studied smooth convex-concave saddle point problem with a bilinear coupling function, which takes the form

$$\min_{x \in \mathbb{R}^{d_x}} \max_{y \in \mathbb{R}^{d_y}} F(x, y) = f(x) + y^\top \mathbf{A}x - g(y), \tag{1}$$

where $f(x)\colon \mathbb{R}^{d_x} \to \mathbb{R}$ and $g(y)\colon \mathbb{R}^{d_y} \to \mathbb{R}$ are smooth and convex functions, and $\mathbf{A} \in \mathbb{R}^{d_y \times d_x}$ is a coupling matrix.

Problem (1) has a large number of application, some of which we now briefly introduce.

### 1.1 Empirical risk minimization

A classical application is the regularized empirical risk minimization (ERM) with linear predictors, which is a classical supervised learning problem. Given a data matrix $\mathbf{A} = [a_1, \ldots, a_n]^\top \in \mathbb{R}^{n \times d}$, where $a_i \in \mathbb{R}^d$ is the feature vector of the $i$-th data entry, our goal is to find a solution of

$$\min_x f(x) + \ell(\mathbf{A}x), \tag{2}$$

where $f(x) : \mathbb{R}^d \to \mathbb{R}$ is a convex regularizer, $\ell(y) : \mathbb{R}^n \to \mathbb{R}$ is a convex loss function, and $x \in \mathbb{R}^d$ is a linear predictor. Alterantively, one can solve the following equivalent saddle-point reformulation

---

[*]King Abdullah University of Science and Technology, Thuwal, Saudi Arabia

[†]Moscow Institute of Physics and Technology, Dolgoprudny, Russia

[‡]Institute for System Programming RAS, Research Center for Trusted Artificial Intelligence, Moscow, Russia

[§]National Research University Higher School of Economics, Moscow, Russia

36th Conference on Neural Information Processing Systems (NeurIPS 2022).

of problem (2):

$$\min_x \max_y f(x) + y^\top \mathbf{A} x - \ell^*(y). \tag{3}$$

The saddle-point reformulation is often preferable. For example, when such a formulation admits a finite sum structure (Zhang and Lin, 2015; Wang and Xiao, 2017), this may reduce the communication complexity in the distributed setting (Xiao et al., 2019), and one may also better exploit the udnerlying sparsity structure (Lei et al., 2017).

## 1.2 Reinforcement learning

In reinforcement learning (RL) we are given a sequence $\{(s_t, a_t, r_t, s_{t+1})\}_{t=1}^n$ generated by a policy $\pi$, where $s_t$ is the state at time step $t$, $a_t$ is the action taken at time step $t$ by policy $\pi$ and $r_t$ is the reward after taking action $a_t$. A key step in many RL algorithms is to estimate the value function of a given policy $\pi$, which is defined as

$$V^\pi(s) = \mathbb{E}\left[\sum_{t=0}^\infty \gamma^t r_t \;\middle|\; s_0 = s, \pi\right], \tag{4}$$

where $\gamma \in (0, 1)$ is a discount factor. A common approach to this problem is to use a linear approximation $V^\pi(s) = \phi(s)^\top x$, where $\phi(s)$ is a feature vector of a state $s$. The model parameter $x$ is often estimated by minimizing the mean squared projected Bellman error

$$\min_x \|\mathbf{A} x - b\|_{\mathbf{C}^{-1}}^2, \tag{5}$$

where $\mathbf{C} = \sum_{t=1}^n \phi(s_t)\phi(s_t)^\top$, $b = \sum_{t=1}^n r_t \phi(s_t)$ and $\mathbf{A} = \mathbf{C} - \gamma \sum_{t=1}^n \phi(s_t)\phi(s_{t+1})^\top$. One can observe that it is hard to apply gradient-based methods to problem (5) because this would require one to compute an inverse of the matrix $\mathbf{C}$. In order to tackle this issue, one can solve an equivalent saddle-point reformulation proposed by Du et al. (2017) instead. This reformulation is given by

$$\min_x \max_y -2y^\top \mathbf{A} x - \|y\|_{\mathbf{C}}^2 + 2b^\top y, \tag{6}$$

and is an instance of problem (1). Solving this reformulation with gradient methods does not require matrix inversion.

## 1.3 Minimization under affine constraints

Next, consider the problem of convex minimization under affine constraints,

$$\min_{\mathbf{A} x = b} f(x), \tag{7}$$

where $b \in \mathrm{range}\mathbf{A}$. This problem covers a wide range of applications, including inverse problems in imaging (Chambolle and Pock, 2016), sketched learning-type applications (Keriven et al., 2018), network flow optimization (Zargham et al., 2013) and optimal transport (Peyré et al., 2019).

Another important application of problem (7) is decentralized distributed optimization (Kovalev et al., 2020; Scaman et al., 2017; Li et al., 2020; Nedic et al., 2017; Arjevani et al., 2020; Ye et al., 2020). In this setting, the distributed minimization problem is often reformulated as

$$\min_{\sqrt{\mathbf{W}}(x_1, \ldots, x_n)^\top = 0} \left[ f(x_1, \ldots, x_n) = \sum_{i=1}^n f_i(x_i) \right], \tag{8}$$

where $f_i(x_i)$ is a function stored locally by a computing node $i \in \{1, \ldots, n\}$ and $\mathbf{W} \in \mathbb{R}^{n \times n}$ is the Laplacian matrix of a graph representing the communication network. The constraint enforces consensus among the nodes: $x_1 = \ldots = x_n$.

One can observe that problem (7) is equivalent to the saddle-point formulation

$$\min_x \max_y f(x) + y^\top \mathbf{A} x - y^\top b, \tag{9}$$

which is another instance of problem (1). State-of-the-art methods often focus on this formulation instead of directly solving (7). In particular, Salim et al. (2021) and Kovalev et al. (2020) obtained optimal algorithms for solving (7) and (8) using this saddle-point approach.

Table 1: Comparison of method (APGD, Algorithm 1) with existing state-of-the-art algorithms for solving problem (1) in the 5 different cases described in section 5.

| | |
|---|---|
| **Strongly-convex-strongly-concave case (section 5.1)** | |
| Algorithm 1 | $\mathcal{O}\left(\max\left\{\sqrt{\frac{L_x}{\mu_x}}, \sqrt{\frac{L_y}{\mu_y}}, \frac{L_{xy}}{\sqrt{\mu_x\mu_y}}\right\}\log\frac{1}{\epsilon}\right)$ |
| Lower bound Zhang et al. (2021b) | $\mathcal{O}\left(\max\left\{\sqrt{\frac{L_x}{\mu_x}}, \sqrt{\frac{L_y}{\mu_y}}, \frac{L_{xy}}{\sqrt{\mu_x\mu_y}}\right\}\log\frac{1}{\epsilon}\right)$ |
| DIPPA Xie et al. (2021) | $\tilde{\mathcal{O}}\left(\max\left\{\sqrt[4]{\frac{L_x^2 L_y}{\mu_x^2\mu_y}}, \sqrt[4]{\frac{L_x L_y^2}{\mu_x\mu_y^2}}, \frac{L_{xy}}{\sqrt{\mu_x\mu_y}}\right\}\log\frac{1}{\epsilon}\right)$ |
| Proximal Best Response Wang and Li (2020) | $\tilde{\mathcal{O}}\left(\max\left\{\sqrt{\frac{L_x}{\mu_x}}, \sqrt{\frac{L_y}{\mu_y}}, \sqrt{\frac{L_{xy}L}{\mu_x\mu_y}}\right\}\log\frac{1}{\epsilon}\right)$ |
| **Affinely constrained minimization case (section 5.2)** | |
| Algorithm 1 | $\mathcal{O}\left(\frac{L_{xy}}{\mu_{xy}}\sqrt{\frac{L_x}{\mu_x}}\log\frac{1}{\epsilon}\right)$ |
| Lower bound Salim et al. (2021) | $\mathcal{O}\left(\frac{L_{xy}}{\mu_{xy}}\sqrt{\frac{L_x}{\mu_x}}\log\frac{1}{\epsilon}\right)$ |
| OPAPC Kovalev et al. (2020) | $\mathcal{O}\left(\frac{L_{xy}}{\mu_{xy}}\sqrt{\frac{L_x}{\mu_x}}\log\frac{1}{\epsilon}\right)$ |
| **Strongly-convex-concave case (section 5.3)** | |
| Algorithm 1 | $\mathcal{O}\left(\max\left\{\frac{\sqrt{L_x L_y}}{\mu_{xy}}, \frac{L_{xy}}{\mu_{xy}}\sqrt{\frac{L_x}{\mu_x}}, \frac{L_{xy}^2}{\mu_{xy}^2}\right\}\log\frac{1}{\epsilon}\right)$ |
| Lower bound | N/A |
| Alt-GDA Zhang et al. (2021a) | $\mathcal{O}\left(\max\left\{\frac{L^2}{\mu_{xy}^2}, \frac{L}{\mu_x}\right\}\log\frac{1}{\epsilon}\right)$ |
| **Bilinear case (section 5.4)** | |
| Algorithm 1 | $\mathcal{O}\left(\frac{L_{xy}^2}{\mu_{xy}^2}\log\frac{1}{\epsilon}\right)$ |
| Lower bound Ibrahim et al. (2020) | $\mathcal{O}\left(\frac{L_{xy}}{\mu_{xy}}\log\frac{1}{\epsilon}\right)$ |
| Azizian et al. (2020) | $\mathcal{O}\left(\frac{L_{xy}}{\mu_{xy}}\log\frac{1}{\epsilon}\right)$ |
| **Convex-concave case (section 5.5)** | |
| Algorithm 1 | $\mathcal{O}\left(\max\left\{\frac{\sqrt{L_x L_y}L_{xy}}{\mu_{xy}^2}, \frac{L_{xy}^2}{\mu_{xy}^2}\right\}\log\frac{1}{\epsilon}\right)$ |
| Lower bound | N/A |

## 1.4 Bilinear min-max problems

Unconstrained bilinear saddle-point problems of the form

$$\min_{x\in\mathbb{R}^{d_x}}\max_{y\in\mathbb{R}^{d_y}} a^\top x + y^\top \mathbf{A}x - b^\top y \tag{10}$$

are another special case of problem (1), one where both $f(x)$ and $g(y)$ are linear functions. While such problems do not usually play an important role in practice, they are often a good testing ground for theoretical purposes (Gidel et al., 2019; Azizian et al., 2020; Zhang et al., 2021a; Mokhtari et al., 2020; Daskalakis et al., 2018; Liang and Stokes, 2019).

## 2 Literature Review and Contributions

In this work we are interested in algorithms able to solve problem (1) with a linear iteration complexity. That is, we are interested in methods that can provably find an $\epsilon$-accurate solution of problem (1) in a number of iterations proportional to $\log\frac{1}{\epsilon}$ (see Definitions 2 and 3). This is typically achieved when

functions $f(x)$ and $g(x)$ are assumed to be strongly convex (see Definition 1). An example of this is the celebrated extragradient method of Korpelevich (1976).

Recent work has shown that linear iteration complexity can be achieved also in the less restrictive case when only one of the functions $f(x)$ and $g(x)$ is strongly convex. This was first shown by Du and Hu (2019), and later improved on by Zhang et al. (2021a).

> *However, and this is the starting point of our research, to the best of our knowledge, there are no algorithms with linear iteration complexity in the case when neither $f(x)$ nor $g(x)$ is strongly convex.*

## 2.1 Acceleration

Loosely speaking, we say that an algorithm is *non-accelerated* if its iteration complexity is proportional to at least the first power of the condition numbers associated with the problem, such as $L_x/\mu_x$ and $L_y/\mu_y$, where $L_x$ and $L_y$ are smoothness constants, and $\mu_x$ and $\mu_y$ are strong convexity constants (see Assumption 1 and Assumption 2). In contrast, the iteration complexity of an *accelerated* algorithm is proportional to the square root of such condition numbers, e.g., $\sqrt{L_x/\mu_x}$ and $\sqrt{L_y/\mu_y}$.

There were several recent attempts to design accelerated algorithms for solving problem (1) (Xie et al., 2021; Wang and Li, 2020; Alkousa et al., 2020). These attempts rely on *stacking multiple algorithms on top of each other*, and result in complicated methods. For example, Lin et al. (2020) use a non-accelerated algorithm as a sub-routine for the inexact accelerated proximal-point method. This approach allows them to obtain accelerated algorithms for solving problem (1) in a straightforward and tractable way. However, this approach has significant drawbacks: the algorithms obtained this way have (i) additional logarithmic factors in their iteration complexity, and (ii) a complex nested structure with the requirement to manually set inner loop sizes, which is a byproduct of the design process based on combining multiple algorithms. This drawback limits the performance of the resulting algorithms in theory, and requires additional fine tuning in practice.

A philosophically different approach to designing such algorithms—one that we adopt in this work—is to attempt to provide a *direct* acceleration of a suitable algorithm for solving problem (1), similarly to what Nesterov (1983) did for convex minimization problems. While this technically more demanding, algorithms obtained this way typically don't have the aforementioned drawbacks. Hence, we follow the latter approach in this work.

## 2.2 Main contributions

In this work we propose an Accelerated Primal-Dual Gradient Method (APDG; Algorithm 1) for solving problem (1) and provide a theoretical analysis of its convergence properties (Theorem 1). In particular, we prove the following results.

(i) When both functions $f(x)$ and $g(y)$ are strongly convex, Algorithm 1 achieves the optimal linear convergence rate, matching the lower bound obtained by Zhang et al. (2021b). To the best of our knowledge, Algorithm 1 is the first optimal algorithm in this regime.

(ii) We establish linear convergence of Algorithm 1 in the case when *only one* of the functions $f(x)$ or $g(y)$ is strongly convex, and $\mathbf{A}$ is a full row or full column rank matrix, respectively. This improves upon the results provided by Du and Hu (2019); Zhang et al. (2021a).

(iii) We establish linear convergence of the Algorithm 1 in the case when *neither* of the functions $f(x)$ nor $g(y)$ is strongly convex, and the matrix $\mathbf{A}$ is square and full rank. To the best of our knowledge, Algorithm 1 is the first algorithm achieving linear convergence in this setting.

Table 1 provides a brief comparison of the complexity of Algorithm 1 (Theorem 1) with the current state of the art. Please refer to section 5 for a detailed discussion of this result and comparison with related work.

## 2.3  General min-max problem and additional contributions

In our work we also consider the saddle-point problem

$$\min_{x \in \mathbb{R}^{d_x}} \max_{y \in \mathbb{R}^{d_y}} F(x, y), \tag{11}$$

where $F(x, y) \colon \mathbb{R}^{d_x} \times \mathbb{R}^{d_y} \to \mathbb{R}$ is a smooth function, which is convex in $x$ and concave in $y$. One can observe that the main problem (1) is a special case of this more general problem (11).

As an additional contribution, we propose a Gradient Descent-Ascent Method with Extrapolation (GDAE) for solving the general convex-concave saddle-point problem (11), and provide a theoretical analysis of its convergence properties.

(i) When the function $F(x, y)$ is strongly convex in $x$ and strongly concave in $y$, GDAE achieves a linear convergence rate, which recovers the convergence result of Cohen et al. (2020).

(ii) Under certain assumptions on the way the variables $x$ and $y$ are coupled by the function $F(x, y)$, we establish linear convergence of GDAE in the case when the function $F(x, y)$ is strongly-convex-concave, convex-strongly-concave, or even just convex-concave. To the best of our knowledge, GDAE is the first algorithm achieving linear convergence under such assumptions.

Please refer to the Appendix for a detailed description of these results and related work.

## 3  Basic Definitions and Assumptions

We start by formalizing the notions of smoothness and strong convexity of a function.

**Definition 1.** *Function $h(z) \colon \mathbb{R}^d \to \mathbb{R}$ is $L$-smooth and $\mu$-strongly convex for $L \geq \mu \geq 0$, if for all $z_1, z_2 \in \mathbb{R}^d$ the following inequality holds:*

$$\frac{\mu}{2}\|z_1 - z_2\|^2 \leq \mathrm{D}_h(z_1, z_2) \leq \frac{L}{2}\|z_1 - z_2\|^2. \tag{12}$$

*Above, $\mathrm{D}_h(z_1, z_2) = h(z_1) - h(z_2) - \langle \nabla h(z_2), z_1 - z_2 \rangle$ is the Bregman divergence associated with the function $h(z)$.*

We are now ready to state the main assumptions that we impose on problem (1). We start with Assumptions 1 and 2 that formalize the strong-convexity and smoothness properties of functions $f(x)$ and $g(y)$.

**Assumption 1.** *Function $f(x)$ is $L_x$-smooth and $\mu_x$-strongly convex for $L_x \geq \mu_x \geq 0$.*

**Assumption 2.** *Function $g(y)$ is $L_y$-smooth and $\mu_y$-strongly convex for $L_y \geq \mu_y \geq 0$.*

Note, that $\mu_x$ and $\mu_y$ are allowed to be zero. That is, both $f(x)$ and $g(y)$ are allowed to be non-strongly convex.

The following assumption formalizes the spectral properties of matrix $\mathbf{A}$.

**Assumption 3.** *There exist constants $L_{xy} > \mu_{xy}, \mu_{yx} \geq 0$ such that*

$$\mu_{xy}^2 \leq \begin{cases} \lambda_{\min}^+(\mathbf{A}\mathbf{A}^\top) & \nabla g(y) \in \mathrm{range}\,\mathbf{A} \text{ for all } y \in \mathbb{R}^{d_y} \\ \lambda_{\min}(\mathbf{A}\mathbf{A}^\top) & \text{otherwise} \end{cases}$$

$$\mu_{yx}^2 \leq \begin{cases} \lambda_{\min}^+(\mathbf{A}^\top\mathbf{A}) & \nabla f(x) \in \mathrm{range}\,\mathbf{A}^\top \text{ for all } x \in \mathbb{R}^{d_x} \\ \lambda_{\min}(\mathbf{A}^\top\mathbf{A}) & \text{otherwise} \end{cases}$$

$$L_{xy}^2 \geq \lambda_{\max}(\mathbf{A}^\top\mathbf{A}) = \lambda_{\max}(\mathbf{A}\mathbf{A}^\top),$$

*where $\lambda_{\min}(\cdot)$, $\lambda_{\min}^+(\cdot)$ and $\lambda_{\max}(\cdot)$ denote the smallest, smallest positive and largest eigenvalue of a matrix, respectively, and* range *denotes the range space of a matrix.*

By $\mathcal{S} \subset \mathbb{R}^{d_x} \times \mathbb{R}^{d_y}$ we denote the solution set of problem (1). Note that $(x^*, y^*) \in \mathcal{S}$ if and only if $(x^*, y^*)$ satisfies the first-order optimality conditions

$$\begin{cases} \nabla_x F(x^*, y^*) = \nabla f(x^*) + \mathbf{A}^\top y^* = 0, \\ \nabla_y F(x^*, y^*) = -\nabla g(y^*) + \mathbf{A}x^* = 0. \end{cases} \tag{13}$$

Our main goal is to propose an algorithm for finding a solution to problem (1). Numerical iterative algorithms typically find an approximate solution of a given problem. We formalize this through the following definition.

**Definition 2.** *Let the solution set $\mathcal{S}$ be nonempty. We call a pair of vectors $(x, y) \in \mathbb{R}^{d_x} \times \mathbb{R}^{d_y}$ an $\epsilon$-accurate solution of problem* (1) *for a given accuracy $\epsilon > 0$ if it satisfies*

$$\min_{(x^*, y^*) \in \mathcal{S}} \max \left\{ \|x - x^*\|^2, \|y - y^*\|^2 \right\} \le \epsilon. \tag{14}$$

We also want to propose an *efficient* algorithm for solving problem (1). That is, we want to propose an algorithm with the the lowest possible *iteration complexity*, which we define next.

**Definition 3.** *The iteration complexity of an algorithm for solving problem* (1) *is the number of iterations the algorithm requires to find an $\epsilon$-accurate solution of this problem. At each iteration the algorithm is allowed to perform $\mathcal{O}(1)$ computations of the gradients $\nabla f(x)$ and $\nabla g(y)$ and matrix-vector multiplications with matrices $\mathbf{A}$ and $\mathbf{A}^\top$.*

## 4    Accelerated Primal-Dual Gradient Method

---
**Algorithm 1** APDG: Accelerated Primal-Dual Gradient Method

---
1: **Input:** $x^0 \in \text{range}\mathbf{A}^\top, y^0 \in \text{range}\mathbf{A}, \eta_x, \eta_y, \alpha_x, \alpha_y, \beta_x, \beta_y > 0, \tau_x, \tau_y, \sigma_x, \sigma_y \in (0, 1], \theta \in (0, 1)$
2: $x_f^0 = x^0$
3: $y_f^0 = y^{-1} = y^0$
4: **for** $k = 0, 1, 2, \ldots$ **do**
5:     $y_m^k = y^k + \theta(y^k - y^{k-1})$
6:     $x_g^k = \tau_x x^k + (1 - \tau_x) x_f^k$
7:     $y_g^k = \tau_y y^k + (1 - \tau_y) y_f^k$
8:     $x^{k+1} = x^k + \eta_x \alpha_x (x_g^k - x^k) - \eta_x \beta_x \mathbf{A}^\top (\mathbf{A}x^k - \nabla g(y_g^k)) - \eta_x \left( \nabla f(x_g^k) + \mathbf{A}^\top y_m^k \right)$
9:     $y^{k+1} = y^k + \eta_y \alpha_y (y_g^k - y^k) - \eta_y \beta_y \mathbf{A}(\mathbf{A}^\top y^k + \nabla f(x_g^k)) - \eta_y (\nabla g(y_g^k) - \mathbf{A}x^{k+1})$
10:     $x_f^{k+1} = x_g^k + \sigma_x (x^{k+1} - x^k)$
11:     $y_f^{k+1} = y_g^k + \sigma_y (y^{k+1} - y^k)$
12: **end for**

---

In this section we present the Accelerated Primal-Dual Gradient Method (APDG; Algorithm 1) for solving problem (1). First, we prove an outline of the key ideas used in the development of this algorithm.

### 4.1    Algorithm development strategy

First, we observe that problem (1) is equivalent to the problem of finding a zero of a sum of two monotone operators, $G_1, G_2 : \mathbb{R}^{d_x} \times \mathbb{R}^{d_y} \to \mathbb{R}^{d_x} \times \mathbb{R}^{d_y}$, defined as

$$G_1 : (x, y) \mapsto (\nabla f(x), \nabla g(y)), \tag{15}$$

$$G_2 : (x, y) \mapsto (\mathbf{A}^\top y, -\mathbf{A}x). \tag{16}$$

Indeed, $G_1(x^*, y^*) + G_2(x^*, y^*) = 0$ is just another way to write the optimality conditions (13).

**The Forward Backward algorithm.**    A natural way to tackle this problem is via *Forward Backward algorithm* (Bauschke and Combettes, 2011), the iterates of which have the form

$$(x^{k+1}, y^{k+1}) = J_{G_2} \left( (x^k, y^k) - G_1(x^k, y^k) \right), \tag{17}$$

where the operator $J_{G_2}$ is the inverse of the operator $I + G_2$, and $I$ is the identity operator. Note that $J_{G_2}$ can be written as $J_{G_2} \colon (x, y) \mapsto (x^+, y^+)$, where $(x^+, y^+) \in \mathbb{R}^{d_x} \times \mathbb{R}^{d_y}$ is a solution of the linear system

$$\begin{cases} x^+ = x - \mathbf{A}^\top y^+ \\ y^+ = y + \mathbf{A}x^+ \end{cases}. \tag{18}$$

**Linear extrapolation step.** Next, notice that the computation of operator $J_{G_2}$ requires solving the linear system (18). This is expensive[5] and has to be done at each iteration of the Forward Backward algorithm. Let us instead consider the related problem

$$\begin{cases} x^+ = x - \mathbf{A}^\top y_m \\ y^+ = y + \mathbf{A}x^+ \end{cases}, \tag{19}$$

where $y_m \in \mathbb{R}^{d_y}$ is a newly introduced variable. It's easy to observe that (19) is equivalent to (18) when $y_m = y^+$. Next, notice that choosing $y_m = y$ makes (19) easy to solve. However, it turns out that the convergence analysis of an algorithm with this approximation may be challenging (Zhang et al., 2021a), especially if we want to combine it with other techniques, such as acceleration. Our key idea is to propose a better alternative: the *linear extrapolation step*

$$y_m = y + \theta(y - y^-), \tag{20}$$

where $y^- \in \mathbb{R}^{d_y}$ corresponds to $y$ obtained from the previous iteration of the Forward Backward algorithm, and $\theta \in (0, 1]$ is an extrapolation parameter. The linear extrapolation step was introduced by Chambolle and Pock (2011) in the analysis of the Primal-Dual Hybrid Gradient algorithm[6].

**Nesterov acceleration.** Next, we note that operator $G_1$ is equal to the gradient of the (potential) function $(x, y) \mapsto f(x) + g(y)$ function. This function is smooth and convex due to Assumptions 1 and 2. This allows us to incorporate the *Nesterov acceleration* mechanism in the Forward Backward algorithm. Nesterov acceleration is known to be a powerful tool which allows to improve convergence properties of gradient methods (Nesterov, 1983, 2003).

## 4.2 Convergence of the algorithm

We are now ready to study the convergence properties of Algorithm 1. We are interested in the case when the following condition holds:

$$\min\left\{\max\left\{\mu_x, \mu_{yx}\right\}, \max\left\{\mu_y, \mu_{xy}\right\}\right\} > 0. \tag{21}$$

In this case one can show that the solution set $\mathcal{S}$ of problem (1) is nonempty. Moreover, *strong duality* holds in this case, as captured by the following lemma.

**Lemma 1.** *Let Assumptions 1, 2 and 3 and condition* (21) *hold. Let $p$ be the optimal value of the primal problem*

$$p = \min_{x \in \mathbb{R}^{d_x}} \left[P(x) = f(x) + g^*(\mathbf{A}x)\right], \tag{22}$$

*and let $d$ be the optimal value of the dual problem*

$$d = \max_{y \in \mathbb{R}^{d_y}} \left[D(y) = -g(y) - f^*(-\mathbf{A}^\top y)\right]. \tag{23}$$

*Then $p = d$ is finite and $(x^*, y^*) \in \mathcal{S}$ if and only if $x^*$ is a solution of the primal problem* (22) *and $y^*$ is a solution of the dual problem* (23).

Under the aforementioned conditions, Algorithm 1 achieves linear convergence. That is, the iteration complexity is proportional to $\log \frac{1}{\epsilon}$.

---

[5]The solution of (18) can be written in a closed form and requires to compute an inverse matrix $(\mathbf{I} + \mathbf{A}^\top \mathbf{A})^{-1}$ or $(\mathbf{I} + \mathbf{A}\mathbf{A}^\top)^{-1}$, where $\mathbf{I}$ is the identity matrix of an appropriate size.

[6]However, the Primal-Dual Hybrid Gradient algorithm is not applicable in our case since it requires to compute the proximal operator of $f(x)$ and $g(y)$ at each iteration. Moreover, Chambolle and Pock (2011) established linear convergence of this algorithm in the strongly-convex-strongly-concave setting only.

**Theorem 1.** *Let Assumptions 1, 2 and 3 and condition* (21) *hold. Then there exist parameters of Algorithm 1 such that its iteration complexity for finding an $\epsilon$-accurate solution of problem* (1) *is*

$$\mathcal{O}\left(\min\left\{T_a, T_b, T_c, T_d\right\}\log\frac{C}{\epsilon}\right), \tag{24}$$

*where $T_a, T_b, T_c, T_d$ are defined as*

$$T_a = \max\left\{\sqrt{\frac{L_x}{\mu_x}}, \sqrt{\frac{L_y}{\mu_y}}, \frac{L_{xy}}{\sqrt{\mu_x\mu_y}}\right\}, \qquad T_b = \max\left\{\frac{\sqrt{L_xL_y}}{\mu_{xy}}, \frac{L_{xy}}{\mu_{xy}}\sqrt{\frac{L_x}{\mu_x}}, \frac{L_{xy}^2}{\mu_{xy}^2}\right\},$$

$$T_c = \max\left\{\frac{\sqrt{L_xL_y}}{\mu_{yx}}, \frac{L_{xy}}{\mu_{yx}}\sqrt{\frac{L_y}{\mu_y}}, \frac{L_{xy}^2}{\mu_{yx}^2}\right\}, \qquad T_d = \max\left\{\frac{\sqrt{L_xL_y}L_{xy}}{\mu_{xy}\mu_{yx}}, \frac{L_{xy}^2}{\mu_{yx}^2}, \frac{L_{xy}^2}{\mu_{xy}^2}\right\},$$

*and $C > 0$ is some constant, which does not depend on $\epsilon$, but possibly depends on $L_x, \mu_x, L_y, \mu_y, L_{xy}, \mu_{xy}, \mu_{yx}$.*

## 5 Discussion of Theorem 1 and Related Work

In this section we comment on the iteration complexity result for Algorithm 1 provided in Theorem 1. We consider important and illustrative special cases of this complexity result and draw connections with the existing results in the literature.

### 5.1 Strongly convex and strongly concave case

In this case $\mu_x, \mu_y > 0$. We can always assume $\mu_{xy} = \mu_{yx} = 0$ in Assumption 3. Then, Algorithm 1 has iteration complexity given by

$$\mathcal{O}\left(\max\left\{\sqrt{\frac{L_x}{\mu_x}}, \sqrt{\frac{L_y}{\mu_y}}, \frac{L_{xy}}{\sqrt{\mu_x\mu_y}}\right\}\log\frac{1}{\epsilon}\right). \tag{25}$$

This improves the current state-of-the-art results

$$\tilde{\mathcal{O}}\left(\max\left\{\sqrt[4]{\frac{L_x^2L_y}{\mu_x^2\mu_y}}, \sqrt[4]{\frac{L_xL_y^2}{\mu_x\mu_y^2}}, \frac{L_{xy}}{\sqrt{\mu_x\mu_y}}\right\}\log\frac{1}{\epsilon}\right) \tag{26}$$

due to Xie et al. (2021), and

$$\tilde{\mathcal{O}}\left(\max\left\{\sqrt{\frac{L_x}{\mu_x}}, \sqrt{\frac{L_y}{\mu_y}}, \sqrt{\frac{L_{xy}L}{\mu_x\mu_y}}\right\}\log\frac{1}{\epsilon}\right), \tag{27}$$

due to Wang and Li (2020), where $\tilde{\mathcal{O}}(\cdot)$ hides additional logarithmic factors, and $L = \max\{L_x, L_y, L_{xy}\}$. Moreover, our result (25) matches the lower complexity bound provided by Zhang et al. (2021b). Hence, *Algorithm 1 is optimal in this regime.*

Apart from our work, algorithms that achieve optimal complexity (25) were developed in three independent works by Thekumparampil et al. (2022); Jin et al. (2022); Du et al. (2022). However, to the best of our knowledge these works were published or appeared on arXiv in 2022, while our work appeared on arXiv in 2021. Hence, Algorithm 1 is the first algorithm which achieves the lower complexity bound (25) for smooth and strongly-convex-strongly-concave saddle-point problems with bilinear coupling.

### 5.2 Affinely-constrainted minimization case

In this case $\mu_x > 0$ and $\mu_y = 0$. Firstly, we consider the case when $L_y = 0$, i.e., $g(y)$ is a linear function. Then, problem (1) is equivalent to the smooth and strongly-convex affinely-constrained minimization problem (7). Algorithm 1 enjoys the linear convergence rate

$$\mathcal{O}\left(\max\left\{\frac{L_{xy}}{\mu_{xy}}\sqrt{\frac{L_x}{\mu_x}}, \frac{L_{xy}^2}{\mu_{xy}^2}\right\}\log\frac{1}{\epsilon}\right), \tag{28}$$

where $\mu_{xy} = \lambda_{\min}^+(\mathbf{A}\mathbf{A}^\top) > 0$ due to Assumption 3. This result recovers the complexity of the APAPC algorithm (Kovalev et al., 2020). It is possible to incorporate the Chebyshev acceleration mechanism (Arioli and Scott, 2014) into Algorithm 1 for solving problem (7) to obtain the improved complexity

$$\mathcal{O}\left(\frac{L_{xy}}{\mu_{xy}}\sqrt{\frac{L_x}{\mu_x}}\log\frac{1}{\epsilon}\right). \tag{29}$$

This matches the complexity of the OPAPC algorithm of Kovalev et al. (2020); Salim et al. (2021), which was shown to be optimal (Salim et al., 2021; Scaman et al., 2017).

### 5.3 Strongly convex and concave case

We also allow $L_y > 0$, i.e., function $g(y)$ is a general, not necessarily linear, smooth and convex function. It is often possible that $\mu_{xy} > 0$ due to Assumption 3; for instance, when $\mathbf{A}$ is a full row rank matrix. Then, Algorithm 1 enjoys the following linear iteration complexity:

$$\mathcal{O}\left(\max\left\{\frac{\sqrt{L_x L_y}}{\mu_{xy}}, \frac{L_{xy}}{\mu_{xy}}\sqrt{\frac{L_x}{\mu_x}}, \frac{L_{xy}^2}{\mu_{xy}^2}\right\}\log\frac{1}{\epsilon}\right). \tag{30}$$

This case was previously studied by Du and Hu (2019); Du et al. (2017); Zhang et al. (2021a). Du and Hu (2019) provided an analysis for an algorithm called Sim-GDA, and established its iteration complexity

$$\mathcal{O}\left(\max\left\{\frac{L_x^3 L_y L_{xy}^2}{\mu_x^2 \mu_{xy}^4}, \frac{L_x^3 L_{xy}^4}{\mu_x^3 \mu_{xy}^4}\right\}\log\frac{1}{\epsilon}\right). \tag{31}$$

This result is substantially worse than our complexity (30); possibly due to a suboptimal analysis. Subsequently, Zhang et al. (2021a) provided an improved analysis for the Sim-GDA algorithm, obtaining the complexity

$$\mathcal{O}\left(\max\left\{\frac{L^3}{\mu_x \mu_{xy}^2}, \frac{L^2}{\mu_x^2}\right\}\log\frac{1}{\epsilon}\right). \tag{32}$$

They also studied the Alt-GDA method, obtaining the complexity

$$\mathcal{O}\left(\max\left\{\frac{L^2}{\mu_{xy}^2}, \frac{L}{\mu_x}\right\}\log\frac{1}{\epsilon}\right), \tag{33}$$

where $L = \max\{L_x, L_y, L_{xy}\}$. However, these results are local, i.e., they are valid only if the initial iterates of these algorithms are close enough to the solution of problem (1). Moreover, these results are still worse than our rate (30) because Sim-GDA and Alt-GDA do not utilize the Nesterov acceleration mechanism, while our Algorithm 1 does.

### 5.4 Bilinear case

In this case $\mu_x = \mu_y = L_x = L_y = 0$. That is, functions $f(x)$ and $g(y)$ are linear. Then, problem (1) turns into the bilinear min-max problem (10), and $\mu_{xy}^2 = \mu_{yx}^2 = \lambda_{\min}^+(\mathbf{A}^\top \mathbf{A}) > 0$ due to Assumption 3. The iteration complexity of Algorithm 1 becomes

$$\mathcal{O}\left(\frac{L_{xy}^2}{\mu_{xy}^2}\log\frac{1}{\epsilon}\right). \tag{34}$$

This recovers the results of Daskalakis et al. (2018); Liang and Stokes (2019); Gidel et al. (2018, 2019); Mishchenko et al. (2020); Mokhtari et al. (2020) for the bilinear min-max problem (10). However, this result is worse than the complexity lower bound

$$\mathcal{O}\left(\frac{L_{xy}}{\mu_{xy}}\log\frac{1}{\epsilon}\right), \tag{35}$$

obtained in the work of Ibrahim et al. (2020), which was reached by Azizian et al. (2020); Du et al. (2022)[7].

---

[7]We provide these results for completeness. The result of Azizian et al. (2020) is better than our result (34) for Algorithm 1 because they specifically focus on solving the bilinear min-max problem (10), while Algorithm 1 aims to solve the much more general convex-concave saddle-point problem (1).

## 5.5 Convex-concave case

In this case $\mu_y = \mu_x = 0$. It is often possible that $\mu_{xy} = \mu_{yx} > 0$ due to Assumption 3, for example, when $\mathbf{A}$ is a square and full rank matrix. Then, the iteration complexity of Algorithm 1 becomes

$$\mathcal{O}\left(\max\left\{\frac{\sqrt{L_x L_y} L_{xy}}{\mu_{xy}^2}, \frac{L_{xy}^2}{\mu_{xy}^2}\right\}\log\frac{1}{\epsilon}\right), \tag{36}$$

which is still linear. This complexity result generalizes the result (34) for bilinear min-max problems as it allows for general, not necessarily linear, convex and smooth functions $f(x)$ and $g(x)$. To the best of our knowledge, Algorithm 1 is the first algorithm which can achieve linear convergence for smooth and non-strongly convex non-strongly concave min-max problems with bilinear coupling.

## Acknowledgements

The work of Alexander Gasnikov was supported by a grant for research centers in the field of artificial intelligence, provided by the Analytical Center for the Government of the Russian Federation in accordance with the subsidy agreement (agreement identifier 000000D730321P5Q0002) and the agreement with the Ivannikov Institute for System Programming of the Russian Academy of Sciences dated November 2, 2021 No. 70-2021-00142.

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
