# Appendix

In Appendix A we present the new GDAE algorithm, in Appendix B we provide a proof of Lemma 1, in Appendix C we provide a proof of Theorem 1, and in Appendix D we provide a proof of Theorem 2.

## A  A Novel Gradient Method for General Convex-Concave Saddle-Point Problems

---

**Algorithm 2** GDAE: Gradient Descent-Ascent with Extrapolation

---

**Input:** $x^0 \in \mathbb{R}^{d_x}$, $y^0 \in \mathbb{R}^{d_y}$, $\eta_x, \eta_y > 0$, $\theta \in (0, 1)$
$x^{-1} = x^0$
$y^{-1} = y^0$
**for** $k = 0, 1, 2, \ldots$ **do**
    $x^{k+1} = x^k - \eta_x \nabla_x F(x^k, y^k) - \eta_x \theta(\nabla_x F(x^{k-1}, y^k) - \nabla_x F(x^{k-1}, y^{k-1}))$
    $y^{k+1} = y^k + \eta_y \nabla_y F(x^{k+1}, y^k)$
**end for**

---

In this section we present a new method—Gradient Descent-Ascent Method with Extrapolation (GDAE; Algorithm 2)—for solving problem (11).

### A.1  Assumptions and definitions

First, we state the main assumptions that we impose on problem (11).

**Assumption 4.** *Function $F(x, y)$ is $L_x$-smooth and $\mu_x$-strongly convex in $x$ and $L_y$-smooth and $\mu_y$-strongly concave in $y$, where $L_x \geq \mu_x \geq 0$, $L_y \geq \mu_y \geq 0$.*

Assumption 4 generalizes the smoothness and strong convexity Assumptions 1 and 2 imposed on problem (1).

**Assumption 5.** *There exists a constant $L_{xy} > 0$ such that for all $x, x_1, x_2 \in \mathbb{R}^{d_x}$ and $y, y_1, y_2 \in \mathbb{R}^{d_y}$, the following inequalities hold:*

$$\begin{aligned} \|\nabla_x F(x, y_1) - \nabla_x F(x, y_2)\| &\leq L_{xy}\|y_1 - y_2\|, \\ \|\nabla_y F(x_1, y) - \nabla_y F(x_2, y)\| &\leq L_{xy}\|x_1 - x_2\|. \end{aligned} \tag{37}$$

**Assumption 6.** *There exist constants $\mu_{xy}, \mu_{yx} \geq 0$ such that for all $x, x_1, x_2 \in \mathbb{R}^{d_x}$ and $y, y_1, y_2 \in \mathbb{R}^{d_y}$, the following inequalities hold:*

$$\begin{aligned} \|\nabla_x F(x, y_1) - \nabla_x F(x, y_2)\| &\geq \mu_{xy}\|y_1 - y_2\|, \\ \|\nabla_y F(x_1, y) - \nabla_y F(x_2, y)\| &\geq \mu_{yx}\|x_1 - x_2\|. \end{aligned} \tag{38}$$

Assumptions 5 and 6 combined form a generalized version of Assumption 3 for problem (11). Indeed, if one assumes that (37) and (38) hold for problem (1), then the following inequalities hold

$$\begin{aligned} \mu_{xy}^2 &\leq \lambda_{\min}(\mathbf{A}\mathbf{A}^\top) \leq L_{xy}^2, \\ \mu_{yx}^2 &\leq \lambda_{\min}(\mathbf{A}^\top\mathbf{A}) \leq L_{xy}^2, \end{aligned} \tag{39}$$

which can be seen as a simplified version of Assumption 3.

Next, we recall several basic definitions. Similarly to section 3, by $\mathcal{S} \subset \mathbb{R}^{d_x} \times \mathbb{R}^{d_y}$ we denote the solution set of problem (11). Note that $(x^*, y^*) \in \mathcal{S}$ if and only if $(x^*, y^*)$ satisfies the optimality conditions

$$\begin{cases} \nabla_x F(x^*, y^*) = 0, \\ \nabla_y F(x^*, y^*) = 0. \end{cases} \tag{40}$$

We also use notions of iteration complexity for achieving an $\epsilon$-accurate solution analogous to Definitions 2 and 3, respectively.

## A.2 Algorithm development

We now present the main ingredients and intuition behind the development of our method.

**Implicit gradient descent-ascent.** First, we recall the iterations of the Forward-Backward algorithm (17), which can be written in the form

$$\begin{cases} x^+ = x - \eta_x \nabla f(x) - \eta_x \mathbf{A}^\top y^+ \\ y^+ = y - \eta_y \nabla g(y) + \eta_y \mathbf{A} x^+ \end{cases}, \tag{41}$$

where $\eta_x, \eta_y > 0$ are stepsizes. Iterations (41) can also be written in terms of the gradients $\nabla_x F(x, y)$ and $\nabla_y F(x, y)$,

$$\begin{cases} x^+ = x - \eta_x \nabla_x F(x, y^+) \\ y^+ = y + \eta_y \nabla_y F(x^+, y) \end{cases}, \tag{42}$$

which makes the method applicable to the general problem (11).

Iterations (42) were the foundation for the development of Algorithm 1 for solving problem (1), with strong convergence properties established by Theorem 1. Hence, we expect that this approach would work for solving the more general problem (11). However, (42) is an implicit algorithm and can't be applied in its current state.

**Gradient extrapolation.** In analogy to the development of Algorithm 1, we want to find a good approximation of the implicit iterations (42). A naive solution would be using the approximation

$$\begin{cases} x^+ = x - \eta_x \nabla_x F(x, y_m) \\ y^+ = y + \eta_y \nabla_y F(x^+, y) \end{cases}, \tag{43}$$

where $y_m \approx y^+$. Similarly to Section 4.1, we could use $y_m = y$, which would lead to the Alt-GDA algorithm (Zhang et al., 2021a), or $y_m = y + \theta(y - y^-)$, which is a linear extrapolation step (Chambolle and Pock, 2011).

The linear extrapolation step with $\theta = 1$ is based on the "assumption" that $y^+ \approx y_m = y + (y - y^-)$, or equivalently, $y^+ - y \approx y - y^-$. We can use a similar intuition for the gradients $\nabla_x F(x, y)$ rather than the iterates $y$. In particular, we "assume" that

$$\nabla_x F(x, y^+) - \nabla_x F(x, y) \approx \nabla_x F(x^-, y) - \nabla_x F(x^-, y^-),$$

or equivalently,

$$\begin{cases} \nabla_x F(x, y^+) \approx \Delta_x \\ \Delta_x = \nabla_x F(x, y) + (\nabla_x F(x^-, y) - \nabla_x F(x^-, y^-)) \end{cases}.$$

This intuition leads to the following novel update rule, which we call *gradient extrapolation step*:

$$\begin{cases} \Delta_x = \nabla_x F(x, y) + \theta(\nabla_x F(x^-, y) - \nabla_x F(x^-, y^-)) \\ x^+ = x - \eta_x \Delta_x \end{cases}.$$

Above, $\theta \in (0, 1]$ is the extrapolation parameter. We use this gradient extrapolation step together with the update rule for $y$ from (42) in the design of our Algorithm 2.

## A.3 Convergence of Algorithm 2 and related work

We now present Theorem 2, which establishes linear convergence rate for Algorithm 2 under Assumptions 4, 5 and 6.

**Theorem 2.** *Let Assumptions 4, 5 and 6 and condition* (21) *hold. Then there exist parameters of Algorithm 2 such that the iteration complexity for finding an $\epsilon$-accurate solution of problem* (11) *is*

$$\mathcal{O}\left(\min\{T_a, T_b, T_c, T_d\} \log \frac{C}{\epsilon}\right), \tag{44}$$

*where* $T_a, T_b, T_c, T_d$ *are defined as*

$$T_a = \max\left\{\frac{L_x}{\mu_x}, \frac{L_y}{\mu_y}, \frac{L_{xy}}{\sqrt{\mu_x \mu_y}}\right\}, \qquad T_b = \max\left\{\frac{L_x}{\mu_x}, \frac{L_x L_y}{\mu_{xy}^2}, \frac{L_{xy}^2}{\mu_{xy}^2}\right\},$$

$$T_c = \max\left\{\frac{L_y}{\mu_y}, \frac{L_x L_y}{\mu_{yx}^2}, \frac{L_{xy}^2}{\mu_{yx}^2}\right\}, \qquad T_d = \max\left\{\frac{L_x L_y}{\mu_{xy}^2}, \frac{L_x L_y}{\mu_{yx}^2}, \frac{L_{xy}^2}{\mu_{xy}^2}, \frac{L_{xy}^2}{\mu_{yx}^2}\right\},$$

*and* $C > 0$ *is some constant, which does not depend on* $\epsilon$, *but possibly depends on* $L_x, \mu_x, L_y, \mu_y, L_{xy}, \mu_{xy}, \mu_{yx}$.

Consider the case when $\mu_x, \mu_y > 0$. In this case the iteration complexity of Algorithm 2 becomes

$$\mathcal{O}\left(\max\left\{\frac{L_x}{\mu_x}, \frac{L_y}{\mu_y}, \frac{L_{xy}}{\sqrt{\mu_x \mu_y}}\right\} \log \frac{1}{\epsilon}\right). \tag{45}$$

This recovers the result of Cohen et al. (2020). Moreover, when $\mu_x = \mu_y$, this result recovers the complexity of solving problem (11) by a number of known algorithms, including the extragradient method (Korpelevich, 1976), optimistic gradient method (Daskalakis et al., 2018; Gidel et al., 2018), and the dual extrapolation method (Nesterov and Scrimali, 2006).

Finally, consider then opposite case when at least one of the constants $\mu_x$ and $\mu_y$ is zero. To the best of our knowledge, there are no algorithms that can achieve a linear convergence. However, Algorithm 2 can still achieve linear iteration complexity provided that condition (21) is satisfied.

## B  Proof of Lemma 1

**Part 1.** Let us first show that primal problem (22) has at least a single solution $x^* \in \mathbb{R}^{d_x}$.

Condition (21) implies that $\max\{\mu_x, \mu_{yx}\} > 0$. If $\mu_x > 0$ then function $P(x)$ is obviously strongly convex and primal problem indeed has a solution. Consider the opposite case $\mu_x = 0$. Then $\mu_{yx} > 0$ due to condition (21).

Assumption 3 and $\mu_{yx} > 0$ imply that $\nabla f(x) \in \text{range} \mathbf{A}^\top$ for all $x \in \mathbb{R}^{d_x}$. Hence,

$$f(x + x') = f(x) \text{ for all } x \in \mathbb{R}^{d_x}, x' \in \ker \mathbf{A}. \tag{46}$$

Using the definition of $P(x)$ we get

$$\begin{aligned} P(x + x') &= f(x + x') + g^*(\mathbf{A}(x + x')) \\ &= f(x) + g^*(\mathbf{A}x) \\ &= P(x) \end{aligned}$$

for all $x \in \mathbb{R}^{d_x}, x' \in \ker \mathbf{A}$. From this one can conclude that

$$\min_{x \in \mathbb{R}^{d_x}} P(x) = \min_{x \in x^0 + \text{range}\mathbf{A}^\top} P(x).$$

for any vector $x^0 \in \mathbb{R}^{d_x}$. From the definition of $P(x)$ it follows that $P(x)$ is $\mu_{yx}$-strongly convex on any affine space $x^0 + \text{range}\mathbf{A}^\top$ for arbitrary $x^0 \in \mathbb{R}^{d_x}$. Hence, problem $\min_{x \in x^0 + \text{range}\mathbf{A}^\top} P(x)$ has a unique solution and primal problem $\min_{x \in \mathbb{R}^{d_x}} P(x)$ has at least a single solution $x^*$.

**Part 2.** Let us show that there exists $y^* \in \mathbb{R}^{d_y}$ such that $(x^*, y^*) \in \mathcal{S}$, i.e., $(x^*, y^*)$ satisfy optimality conditions (13).

Let us show that $-\nabla f(x^*) \in \mathbf{A}^\top \partial g^*(\mathbf{A}x^*)$. We use condition (21) which implies $\max\{\mu_y, \mu_{xy}\} > 0$. If $\mu_y > 0$, then function $g^*(y)$ is smooth and our statement is trivial. Consider the opposite case $\mu_y = 0$. Then $\mu_{xy} > 0$ due to condition (21).

Assumption 3 and $\mu_{xy} > 0$ imply that $\nabla g(y) \in \text{range}\mathbf{A}$ for all $y \in \mathbb{R}^{d_y}$. Hence, $\text{dom } g^*(\cdot) \subset \text{range}\mathbf{A}$. Let $h(x) = g^*(\mathbf{A}x)$. From standard theory it follows that $-\nabla f(x^*) \in \partial h(x^*)$ or

$$h(x) \geq h(x^*) - \langle \nabla f(x^*), x - x^* \rangle \text{ for all } x \in \mathbb{R}^{d_x},$$

From this one can conclude that

$$\langle \nabla f(x^*), x - x^* \rangle \geq 0 \text{ for all } x \in x^* + \ker \mathbf{A},$$

which implies $\nabla f(x^*) \in (\ker \mathbf{A})^\perp = \operatorname{range} \mathbf{A}^\top$. Hence, there exists vector $y^* \in \mathbb{R}^{d_y}$ such that $-\nabla f(x^*) = \mathbf{A}^\top y^*$. Now, we can write

$$h(x) \geq h(x^*) + \langle \mathbf{A}^\top y^*, x - x^* \rangle \text{ for all } x \in \mathbb{R}^{d_x},$$

which is equivalent to

$$g^*(\mathbf{A}x) \geq g^*(\mathbf{A}x^*) + \langle y^*, \mathbf{A}x - \mathbf{A}x^* \rangle \text{ for all } x \in \mathbb{R}^{d_x}.$$

The latter can be written as

$$g^*(y) \geq g^*(\mathbf{A}x^*) + \langle y^*, y - \mathbf{A}x^* \rangle \text{ for all } y \in \operatorname{range} \mathbf{A}.$$

But $\operatorname{dom} g^*(\cdot) \subset \operatorname{range} \mathbf{A}$, which means that $g^*(y) = +\infty$ for all $y \notin \operatorname{range} \mathbf{A}$. This implies

$$g^*(y) \geq g^*(\mathbf{A}x^*) + \langle y^*, y - \mathbf{A}x^* \rangle \text{ for all } y \in \mathbb{R}^{d_y},$$

which is a definition of $y^* \in \partial g^*(\mathbf{A}x^*)$. An equivalent for this is $\nabla g(y^*) = \mathbf{A}x^*$, which together with $-\nabla f(x^*) = \mathbf{A}^\top y^*$ form optimality condition (13).

**Part 3.** We showed that there exists a pair of vectors $(x^*, y^*) \in \mathbb{R}^{d_x} \times \mathbb{R}^{d_y}$ which is a saddle point of the function $F(x, y)$ in problem (1). Hence, strong duality holds and proof of the rest of Lemma 1 is trivial. □

## C Proof of Theorem 1

**Lemma 2.** *There exists a solution $(x^*, y^*) \in \mathcal{S}$ of the problem (1) such that for all $k = 0, 1, 2, \ldots$ the iterates of Algorithm 1 satisfy*

$$\|\mathbf{A}(x^k - x^*)\| \geq \mu_{yx} \|x^k - x^*\|,$$
$$\|\mathbf{A}^\top(y^k - y^*)\| \geq \mu_{xy} \|y^k - y^*\|. \tag{47}$$

*Proof.* The proof of this lemma is a trivial extension of the derivations from the proof of Lemma 1. □

**Lemma 3.** *Let $\tau_x$ be defined as*

$$\tau_x = (\sigma_x^{-1} + 1/2)^{-1}. \tag{48}$$

*Let $\alpha_x$ be defined as*

$$\alpha_x = \mu_x. \tag{49}$$

*Let $\beta_x$ be defined as*

$$\beta_x = \min\left\{ \frac{1}{2L_y}, \frac{1}{2\eta_x L_{xy}^2} \right\}. \tag{50}$$

*Then, the following inequality holds:*

$$\frac{1}{\eta_x} \|x^{k+1} - x^*\|^2 \leq \left( \frac{1}{\eta_x} - \mu_x - \beta_x \mu_{yx}^2 \right) \|x^k - x^*\|^2 + \left( \mu_x + L_x \sigma_x - \frac{1}{2\eta_x} \right) \|x^{k+1} - x^k\|^2$$
$$+ \mathrm{D}_g(y_g^k, y^*) - \mathrm{D}_f(x_g^k, x^*) - \frac{2}{\sigma_x} \mathrm{D}_f(x_f^{k+1}, x^*) + \left( \frac{2}{\sigma_x} - 1 \right) \mathrm{D}_f(x_f^k, x^*)$$
$$- 2\langle \mathbf{A}^\top(y_m^k - y^*), x^{k+1} - x^* \rangle. \tag{51}$$

*Proof.* Using Line 8 of the Algorithm 1 we get

$$\frac{1}{\eta_x} \|x^{k+1} - x^*\|^2 = \frac{1}{\eta_x} \|x^k - x^*\|^2 + \frac{2}{\eta_x} \langle x^{k+1} - x^k, x^{k+1} - x^* \rangle - \frac{1}{\eta_x} \|x^{k+1} - x^k\|^2$$
$$= \frac{1}{\eta_x} \|x^k - x^*\|^2 + 2\alpha_x \langle x_g^k - x^k, x^{k+1} - x^* \rangle - 2\beta_x \langle \mathbf{A}^\top(\mathbf{A}x^k - \nabla g(y_g^k)), x^{k+1} - x^* \rangle$$
$$- 2\langle \nabla f(x_g^k) + \mathbf{A}^\top y_m^k, x^{k+1} - x^* \rangle - \frac{1}{\eta_x} \|x^{k+1} - x^k\|^2.$$

Using the parallelogram rule we get

$$\frac{1}{\eta_x} \|x^{k+1} - x^*\|^2 = \frac{1}{\eta_x} \|x^k - x^*\|^2 + \alpha_x \left( \|x_g^k - x^*\|^2 - \|x_g^k - x^{k+1}\|^2 - \|x^k - x^*\|^2 + \|x^{k+1} - x^k\|^2 \right)$$
$$- 2\beta_x \langle \mathbf{A}x^k - \nabla g(y_g^k), \mathbf{A}(x^{k+1} - x^*) \rangle - 2\langle \nabla f(x_g^k) + \mathbf{A}^\top y_m^k, x^{k+1} - x^* \rangle - \frac{1}{\eta_x} \|x^{k+1} - x^k\|^2.$$

Using the optimality condition $\nabla g(y^*) = \mathbf{A}x^*$, which follows from (13), and the parallelogram rule we get

$$\frac{1}{\eta_x} \|x^{k+1} - x^*\|^2 = \frac{1}{\eta_x} \|x^k - x^*\|^2 + \alpha_x \left( \|x_g^k - x^*\|^2 - \|x_g^k - x^{k+1}\|^2 - \|x^k - x^*\|^2 + \|x^{k+1} - x^k\|^2 \right)$$
$$+ \beta_x \left( \|\mathbf{A}(x^{k+1} - x^k)\|^2 - \|\mathbf{A}(x^k - x^*)\|^2 + \|\nabla g(y_g^k) - \nabla g(y^*)\|^2 - \|\nabla g(y_g^k) - \mathbf{A}(x^{k+1})\|^2 \right)$$
$$- 2\langle \nabla f(x_g^k) + \mathbf{A}^\top y_m^k, x^{k+1} - x^* \rangle - \frac{1}{\eta_x} \|x^{k+1} - x^k\|^2.$$

Using Assumption 3, equation 47 and $L_y$-smoothness of $g$ we get

$$\frac{1}{\eta_x}\|x^{k+1} - x^*\|^2 \le \frac{1}{\eta_x}\|x^k - x^*\|^2 + \alpha_x\|x_g^k - x^*\|^2 - \alpha_x\|x^k - x^*\|^2 + \alpha_x\|x^{k+1} - x^k\|^2$$
$$+ \beta_x L_{xy}^2\|x^{k+1} - x^k\|^2 - \beta_x \mu_{yx}^2\|x^k - x^*\|^2 + 2\beta_x L_y D_g(y_g^k, y^*)$$
$$- 2\langle \nabla f(x_g^k) + \mathbf{A}^\top y_m^k, x^{k+1} - x^*\rangle - \frac{1}{\eta_x}\|x^{k+1} - x^k\|^2$$
$$= \left(\frac{1}{\eta_x} - \alpha_x - \beta_x \mu_{yx}^2\right)\|x^k - x^*\|^2 + \left(\beta_x L_{xy}^2 + \alpha_x - \frac{1}{\eta_x}\right)\|x^{k+1} - x^k\|^2$$
$$+ 2\beta_x L_y D_g(y_g^k, y^*) + \alpha_x\|x_g^k - x^*\|^2 - 2\langle \nabla f(x_g^k) + \mathbf{A}^\top y_m^k, x^{k+1} - x^*\rangle.$$

Using the optimality condition $\nabla f(x^*) + \mathbf{A}^\top y^* = 0$, which follows from (13), we get

$$\frac{1}{\eta_x}\|x^{k+1} - x^*\|^2 \le \left(\frac{1}{\eta_x} - \alpha_x - \beta_x \mu_{yx}^2\right)\|x^k - x^*\|^2 + \left(\beta_x L_{xy}^2 + \alpha_x - \frac{1}{\eta_x}\right)\|x^{k+1} - x^k\|^2 + 2\beta_x L_y D_g(y_g^k, y^*)$$
$$+ \alpha_x\|x_g^k - x^*\|^2 - 2\langle \nabla f(x_g^k) - \nabla f(x^*), x^{k+1} - x^*\rangle - 2\langle \mathbf{A}^\top(y_m^k - y^*), x^{k+1} - x^*\rangle$$
$$= \left(\frac{1}{\eta_x} - \alpha_x - \beta_x \mu_{yx}^2\right)\|x^k - x^*\|^2 + \left(\beta_x L_{xy}^2 + \alpha_x - \frac{1}{\eta_x}\right)\|x^{k+1} - x^k\|^2$$
$$+ 2\beta_x L_y D_g(y_g^k, y^*) + \alpha_x\|x_g^k - x^*\|^2 - 2\langle \nabla f(x_g^k) - \nabla f(x^*), x^{k+1} - x^k + x^k - x_g^k + x_g^k - x^*\rangle$$
$$- 2\langle \mathbf{A}^\top(y_m^k - y^*), x^{k+1} - x^*\rangle$$

Using $\mu_y$-strong convexity of $f$ and Lines 6 and 10 of the Algorithm 1 we get

$$\frac{1}{\eta_x}\|x^{k+1} - x^*\|^2 \le \left(\frac{1}{\eta_x} - \alpha_x - \beta_x \mu_{yx}^2\right)\|x^k - x^*\|^2 + \left(\beta_x L_{xy}^2 + \alpha_x - \frac{1}{\eta_x}\right)\|x^{k+1} - x^k\|^2 + 2\beta_x L_y D_g(y_g^k, y^*)$$
$$+ \alpha_x\|x_g^k - x^*\|^2 - \frac{2}{\sigma_x}\langle \nabla f(x_g^k) - \nabla f(x^*), x_f^{k+1} - x_g^k\rangle + \frac{2(1-\tau_x)}{\tau_x}\langle \nabla f(x_g^k) - \nabla f(x^*), x_f^k - x_g^k\rangle$$
$$- 2D_f(x_g^k, x^*) - \mu_x\|x_g^k - x^*\|^2 - 2\langle \mathbf{A}^\top(y_m^k - y^*), x^{k+1} - x^*\rangle$$
$$= \left(\frac{1}{\eta_x} - \alpha_x - \beta_x \mu_{yx}^2\right)\|x^k - x^*\|^2 + \left(\beta_x L_{xy}^2 + \alpha_x - \frac{1}{\eta_x}\right)\|x^{k+1} - x^k\|^2 + (\alpha_x - \mu_x)\|x_g^k - x^*\|^2$$
$$+ 2\beta_x L_y D_g(y_g^k, y^*) - 2D_f(x_g^k, x^*) - \frac{2}{\sigma_x}\langle \nabla f(x_g^k) - \nabla f(x^*), x_f^{k+1} - x_g^k\rangle$$
$$+ \frac{2(1-\tau_x)}{\tau_x}\langle \nabla f(x_g^k) - \nabla f(x^*), x_f^k - x_g^k\rangle - 2\langle \mathbf{A}^\top(y_m^k - y^*), x^{k+1} - x^*\rangle.$$

Using convexity of $D_f(x, x^*)$ with respect to $x$, which follows from the convexity of $f$, we get

$$\frac{1}{\eta_x}\|x^{k+1} - x^*\|^2 \le \left(\frac{1}{\eta_x} - \alpha_x - \beta_x \mu_{yx}^2\right)\|x^k - x^*\|^2 + \left(\beta_x L_{xy}^2 + \alpha_x - \frac{1}{\eta_x}\right)\|x^{k+1} - x^k\|^2 + (\alpha_x - \mu_x)\|x_g^k - x^*\|^2$$
$$+ 2\beta_x L_y D_g(y_g^k, y^*) - 2D_f(x_g^k, x^*) - \frac{2}{\sigma_x}\langle \nabla f(x_g^k) - \nabla f(x^*), x_f^{k+1} - x_g^k\rangle$$
$$+ \frac{2(1-\tau_x)}{\tau_x}\left(D_f(x_f^k, x^*) - D_f(x_g^k, x^*)\right) - 2\langle \mathbf{A}^\top(y_m^k - y^*), x^{k+1} - x^*\rangle.$$

Using $L_x$-smoothness of $D_f(x, x^*)$ with respect to $x$, which follows from the $L_x$-smoothness of $f$, we get

$$\frac{1}{\eta_x}\|x^{k+1} - x^*\|^2 \le \left(\frac{1}{\eta_x} - \alpha_x - \beta_x \mu_{yx}^2\right)\|x^k - x^*\|^2 + \left(\beta_x L_{xy}^2 + \alpha_x - \frac{1}{\eta_x}\right)\|x^{k+1} - x^k\|^2 + (\alpha_x - \mu_x)\|x_g^k - x^*\|^2$$
$$+ 2\beta_x L_y D_g(y_g^k, y^*) - 2D_f(x_g^k, x^*) - \frac{2}{\sigma_x}\left(D_f(x_f^{k+1}, x^*) - D_f(x_g^k, x^*) - \frac{L_x}{2}\|x_f^{k+1} - x_g^k\|^2\right)$$
$$+ \frac{2(1-\tau_x)}{\tau_x}\left(D_f(x_f^k, x^*) - D_f(x_g^k, x^*)\right) - 2\langle \mathbf{A}^\top(y_m^k - y^*), x^{k+1} - x^*\rangle.$$

Using Line 10 of the Algorithm 1 we get

$$
\frac{1}{\eta_x}\|x^{k+1}-x^*\|^2 \leq \left(\frac{1}{\eta_x}-\alpha_x-\beta_x\mu_{yx}^2\right)\|x^k-x^*\|^2 + \left(\beta_x L_{xy}^2+\alpha_x-\frac{1}{\eta_x}\right)\|x^{k+1}-x^k\|^2 + (\alpha_x-\mu_x)\|x_g^k-x^*\|^2
$$

$$
+ 2\beta_x L_y \mathrm{D}_g(y_g^k,y^*) - 2\mathrm{D}_f(x_g^k,x^*) - \frac{2}{\sigma_x}\left(\mathrm{D}_f(x_f^{k+1},x^*) - \mathrm{D}_f(x_g^k,x^*) - \frac{L_x\sigma_x^2}{2}\|x^{k+1}-x^k\|^2\right)
$$

$$
+ \frac{2(1-\tau_x)}{\tau_x}\left(\mathrm{D}_f(x_f^k,x^*)-\mathrm{D}_f(x_g^k,x^*)\right) - 2\langle \mathbf{A}^\top(y_m^k-y^*),x^{k+1}-x^*\rangle
$$

$$
= \left(\frac{1}{\eta_x}-\alpha_x-\beta_x\mu_{yx}^2\right)\|x^k-x^*\|^2 + \left(\beta_x L_{xy}^2+\alpha_x+L_x\sigma_x-\frac{1}{\eta_x}\right)\|x^{k+1}-x^k\|^2
$$

$$
+ (\alpha_x-\mu_x)\|x_g^k-x^*\|^2 + 2\beta_x L_y \mathrm{D}_g(y_g^k,y^*) + \left(\frac{2}{\sigma_x}-\frac{2}{\tau_x}\right)\mathrm{D}_f(x_g^k,x^*) - \frac{2}{\sigma_x}\mathrm{D}_f(x_f^{k+1},x^*)
$$

$$
+ \left(\frac{2}{\tau_x}-2\right)\mathrm{D}_f(x_f^k,x^*) - 2\langle \mathbf{A}^\top(y_m^k-y^*),x^{k+1}-x^*\rangle.
$$

Using the definition of $\tau_x$, $\alpha_x$ and $\beta_x$ we get

$$
\frac{1}{\eta_x}\|x^{k+1}-x^*\|^2 \leq \left(\frac{1}{\eta_x}-\mu_x-\beta_x\mu_{yx}^2\right)\|x^k-x^*\|^2 + \left(\mu_x+L_x\sigma_x-\frac{1}{2\eta_x}\right)\|x^{k+1}-x^k\|^2
$$

$$
+ \mathrm{D}_g(y_g^k,y^*)-\mathrm{D}_f(x_g^k,x^*) - \frac{2}{\sigma_x}\mathrm{D}_f(x_f^{k+1},x^*) + \left(\frac{2}{\sigma_x}-1\right)\mathrm{D}_f(x_f^k,x^*)
$$

$$
- 2\langle \mathbf{A}^\top(y_m^k-y^*),x^{k+1}-x^*\rangle.
$$

$\square$

**Lemma 4.** *Let $\tau_y$ be defined as*

$$
\tau_y = (\sigma_y^{-1}+1/2)^{-1}. \tag{52}
$$

*Let $\alpha_y$ be defined as*

$$
\alpha_y = \mu_y. \tag{53}
$$

*Let $\beta_y$ be defined as*

$$
\beta_y = \min\left\{\frac{1}{2L_x},\frac{1}{2\eta_y L_{xy}^2}\right\}. \tag{54}
$$

*Then, the following inequality holds:*

$$
\frac{1}{\eta_y}\|y^{k+1}-y^*\|^2 \leq \left(\frac{1}{\eta_y}-\mu_y-\beta_y\mu_{xy}^2\right)\|y^k-y^*\|^2 + \left(\mu_y+L_y\sigma_y-\frac{1}{2\eta_y}\right)\|y^{k+1}-y^k\|^2
$$

$$
+ \mathrm{D}_f(x_g^k,x^*)-\mathrm{D}_g(y_g^k,y^*) - \frac{2}{\sigma_y}\mathrm{D}_g(y_f^{k+1},y^*) + \left(\frac{2}{\sigma_y}-1\right)\mathrm{D}_g(y_f^k,y^*)
$$

$$
+ 2\langle \mathbf{A}(x^{k+1}-x^*),y^{k+1}-y^*\rangle. \tag{55}
$$

*Proof.* The proof is similar to the proof of the previous lemma. $\square$

**Lemma 5.** *Let $\eta_x$ be defined as*

$$
\eta_x = \min\left\{\frac{1}{4(\mu_x+L_x\sigma_x)},\frac{\delta}{4L_{xy}}\right\}, \tag{56}
$$

*and let $\eta_y$ be defined as*

$$
\eta_y = \min\left\{\frac{1}{4(\mu_y+L_y\sigma_y)},\frac{1}{4L_{xy}\delta}\right\}, \tag{57}
$$

*where $\delta > 0$ is a parameter. Let $\theta$ be defined as*

$$
\theta = \theta(\delta,\sigma_x,\sigma_y) = 1 - \max\left\{\rho_a(\delta,\sigma_x,\sigma_y),\rho_b(\delta,\sigma_x,\sigma_y),\rho_c(\delta,\sigma_x,\sigma_y),\rho_d(\delta,\sigma_x,\sigma_y)\right\}, \tag{58}
$$

*where $\rho_a(\delta, \sigma_x, \sigma_y), \rho_b(\delta, \sigma_x, \sigma_y), \rho_c(\delta, \sigma_x, \sigma_y), \rho_d(\delta, \sigma_x, \sigma_y)$ are defined as*

$$\rho_a(\delta, \sigma_x, \sigma_y) = \left[\max\left\{\frac{4(\mu_x + L_x\sigma_x)}{\mu_x}, \frac{2}{\sigma_x}, \frac{4(\mu_y + L_y\sigma_y)}{\mu_y}, \frac{2}{\sigma_y}, \frac{4L_{xy}}{\mu_x\delta}, \frac{4L_{xy}\delta}{\mu_y}\right\}\right]^{-1}, \tag{59}$$

$$\rho_b(\delta, \sigma_x, \sigma_y) = \left[\max\left\{\frac{4(\mu_x + L_x\sigma_x)}{\mu_x}, \frac{2}{\sigma_x}, \frac{8L_x(\mu_y + L_y\sigma_y)}{\mu_{xy}^2}, \frac{2}{\sigma_y}, \frac{2L_{xy}^2}{\mu_{xy}^2}, \frac{8L_xL_{xy}\delta}{\mu_{xy}^2}, \frac{4L_{xy}}{\mu_x\delta}\right\}\right]^{-1}, \tag{60}$$

$$\rho_c(\delta, \sigma_x, \sigma_y) = \left[\max\left\{\frac{4(\mu_y + L_y\sigma_y)}{\mu_y}, \frac{2}{\sigma_y}, \frac{8L_y(\mu_x + L_x\sigma_x)}{\mu_{yx}^2}, \frac{2}{\sigma_x}, \frac{2L_{xy}^2}{\mu_{yx}^2}, \frac{8L_yL_{xy}}{\mu_{yx}^2\delta}, \frac{4L_{xy}\delta}{\mu_y}\right\}\right]^{-1}, \tag{61}$$

$$\rho_d(\delta, \sigma_x, \sigma_y) = \left[\max\left\{\frac{8L_y(\mu_x + L_x\sigma_x)}{\mu_{yx}^2}, \frac{2}{\sigma_x}, \frac{8L_x(\mu_y + L_y\sigma_y)}{\mu_{xy}^2}, \frac{2}{\sigma_y}, \frac{8L_yL_{xy}}{\delta\mu_{yx}^2}, \frac{8L_xL_{xy}\delta}{\mu_{xy}^2}, \frac{2L_{xy}^2}{\mu_{yx}^2}, \frac{2L_{xy}^2}{\mu_{xy}^2}\right\}\right]^{-1}. \tag{62}$$

*Let $\Psi^k$ be the following Lyapunov function:*

$$\Psi^k = \frac{1}{\eta_x}\|x^k - x^*\|^2 + \frac{1}{\eta_y}\|y^k - y^*\|^2 + \frac{2}{\sigma_x}D_f(x_f^k, x^*) + \frac{2}{\sigma_y}D_g(y_f^k, y^*)$$
$$+ \frac{1}{4\eta_y}\|y^k - y^{k-1}\|^2 - 2\langle y^k - y^{k-1}, \mathbf{A}(x^k - x^*)\rangle. \tag{63}$$

*Then, the following inequalities hold*

$$\Psi^k \geq \frac{3}{4\eta_x}\|x^k - x^*\|^2 + \frac{1}{\eta_y}\|y^k - y^*\|^2, \tag{64}$$

$$\Psi^{k+1} \leq \theta\Psi^k. \tag{65}$$

*Proof.* After adding up (51) and (55) we get

$$(\text{LHS}) \leq \left(\frac{1}{\eta_x} - \mu_x - \beta_x\mu_{yx}^2\right)\|x^k - x^*\|^2 + \left(\frac{1}{\eta_y} - \mu_y - \beta_y\mu_{xy}^2\right)\|y^k - y^*\|^2$$
$$+ \left(\mu_x + L_x\sigma_x - \frac{1}{2\eta_x}\right)\|x^{k+1} - x^k\|^2 + \left(\mu_y + L_y\sigma_y - \frac{1}{2\eta_y}\right)\|y^{k+1} - y^k\|^2$$
$$+ \left(\frac{2}{\sigma_x} - 1\right)D_f(x_f^k, x^*) + \left(\frac{2}{\sigma_y} - 1\right)D_g(y_f^k, y^*) + 2\langle y^{k+1} - y_m^k, \mathbf{A}(x^{k+1} - x^*)\rangle.$$

where (LHS) is given as

$$(\text{LHS}) = \frac{1}{\eta_x}\|x^{k+1} - x^*\|^2 + \frac{1}{\eta_y}\|y^{k+1} - y^*\|^2 + \frac{2}{\sigma_x}D_f(x_f^{k+1}, x^*) + \frac{2}{\sigma_y}D_g(y_f^{k+1}, y^*).$$

Using Line 5 of the Algorithm 1 and Assumption 3 we get

$$(\text{LHS}) \leq \left(\frac{1}{\eta_x} - \mu_x - \beta_x\mu_{yx}^2\right)\|x^k - x^*\|^2 + \left(\frac{1}{\eta_y} - \mu_y - \beta_y\mu_{xy}^2\right)\|y^k - y^*\|^2$$
$$+ \left(\mu_x + L_x\sigma_x - \frac{1}{2\eta_x}\right)\|x^{k+1} - x^k\|^2 + \left(\mu_y + L_y\sigma_y - \frac{1}{2\eta_y}\right)\|y^{k+1} - y^k\|^2$$
$$+ \left(\frac{2}{\sigma_x} - 1\right)D_f(x_f^k, x^*) + \left(\frac{2}{\sigma_y} - 1\right)D_g(y_f^k, y^*)$$
$$+ 2\langle y^{k+1} - y^k, \mathbf{A}(x^{k+1} - x^*)\rangle - 2\theta\langle y^k - y^{k-1}, \mathbf{A}(x^{k+1} - x^*)\rangle$$
$$\leq \left(\frac{1}{\eta_x} - \mu_x - \beta_x\mu_{yx}^2\right)\|x^k - x^*\|^2 + \left(\frac{1}{\eta_y} - \mu_y - \beta_y\mu_{xy}^2\right)\|y^k - y^*\|^2$$

$$+ \left( \mu_x + L_x \sigma_x - \frac{1}{2\eta_x} \right) \|x^{k+1} - x^k\|^2 + \left( \mu_y + L_y \sigma_y - \frac{1}{2\eta_y} \right) \|y^{k+1} - y^k\|^2$$

$$+ \left( \frac{2}{\sigma_x} - 1 \right) \mathrm{D}_f(x_f^k, x^*) + \left( \frac{2}{\sigma_y} - 1 \right) \mathrm{D}_g(y_f^k, y^*)$$

$$+ 2\langle y^{k+1} - y^k, \mathbf{A}(x^{k+1} - x^*)\rangle - 2\theta \langle y^k - y^{k-1}, \mathbf{A}(x^k - x^*)\rangle + 2\theta L_{xy} \|y^k - y^{k-1}\| \|x^{k+1} - x^k\|.$$

Using the definition of $\eta_x$ and $\eta_y$ and the fact that $\theta < 1$ we get

$$(\text{LHS}) \leq \left( \frac{1}{\eta_x} - \mu_x - \beta_x \mu_{yx}^2 \right) \|x^k - x^*\|^2 + \left( \frac{1}{\eta_y} - \mu_y - \beta_y \mu_{xy}^2 \right) \|y^k - y^*\|^2$$

$$- \frac{1}{4\eta_x} \|x^{k+1} - x^k\|^2 - \frac{1}{4\eta_y} \|y^{k+1} - y^k\|^2 + \left( \frac{2}{\sigma_x} - 1 \right) \mathrm{D}_f(x_f^k, x^*) + \left( \frac{2}{\sigma_y} - 1 \right) \mathrm{D}_g(y_f^k, y^*)$$

$$+ 2\langle y^{k+1} - y^k, \mathbf{A}(x^{k+1} - x^*)\rangle - 2\theta \langle y^k - y^{k-1}, \mathbf{A}(x^k - x^*)\rangle + \frac{\theta}{2\sqrt{\eta_x \eta_y}} \|y^k - y^{k-1}\| \|x^{k+1} - x^k\|$$

$$\leq \left( \frac{1}{\eta_x} - \mu_x - \beta_x \mu_{yx}^2 \right) \|x^k - x^*\|^2 + \left( \frac{1}{\eta_y} - \mu_y - \beta_y \mu_{xy}^2 \right) \|y^k - y^*\|^2$$

$$- \frac{1}{4\eta_x} \|x^{k+1} - x^k\|^2 - \frac{1}{4\eta_y} \|y^{k+1} - y^k\|^2 + \left( \frac{2}{\sigma_x} - 1 \right) \mathrm{D}_f(x_f^k, x^*) + \left( \frac{2}{\sigma_y} - 1 \right) \mathrm{D}_g(y_f^k, y^*)$$

$$+ 2\langle y^{k+1} - y^k, \mathbf{A}(x^{k+1} - x^*)\rangle - 2\theta \langle y^k - y^{k-1}, \mathbf{A}(x^k - x^*)\rangle + \frac{\theta}{4\eta_x} \|x^{k+1} - x^k\|^2 + \frac{\theta}{4\eta_y} \|y^k - y^{k-1}\|^2$$

$$\leq \left( \frac{1}{\eta_x} - \mu_x - \beta_x \mu_{yx}^2 \right) \|x^k - x^*\|^2 + \left( \frac{1}{\eta_y} - \mu_y - \beta_y \mu_{xy}^2 \right) \|y^k - y^*\|^2$$

$$+ \frac{\theta}{4\eta_y} \|y^k - y^{k-1}\|^2 - \frac{1}{4\eta_y} \|y^{k+1} - y^k\|^2 + \left( \frac{2}{\sigma_x} - 1 \right) \mathrm{D}_f(x_f^k, x^*) + \left( \frac{2}{\sigma_y} - 1 \right) \mathrm{D}_g(y_f^k, y^*)$$

$$+ 2\langle y^{k+1} - y^k, \mathbf{A}(x^{k+1} - x^*)\rangle - 2\theta \langle y^k - y^{k-1}, \mathbf{A}(x^k - x^*)\rangle.$$

Using the definition of $\beta_x$ and $\beta_y$ we get

$$(\text{LHS}) \leq \left( 1 - \eta_x \mu_x - \min\left\{ \frac{\eta_x \mu_{yx}^2}{2L_y}, \frac{\mu_{yx}^2}{2L_{xy}^2} \right\} \right) \frac{1}{\eta_x} \|x^k - x^*\|^2 + \left( 1 - \eta_y \mu_y - \min\left\{ \frac{\eta_y \mu_{xy}^2}{2L_x}, \frac{\mu_{xy}^2}{2L_{xy}^2} \right\} \right) \frac{1}{\eta_y} \|y^k - y^*\|^2$$

$$+ \frac{\theta}{4\eta_y} \|y^k - y^{k-1}\|^2 - \frac{1}{4\eta_y} \|y^{k+1} - y^k\|^2 + \left( \frac{2}{\sigma_x} - 1 \right) \mathrm{D}_f(x_f^k, x^*) + \left( \frac{2}{\sigma_y} - 1 \right) \mathrm{D}_g(y_f^k, y^*)$$

$$+ 2\langle y^{k+1} - y^k, \mathbf{A}(x^{k+1} - x^*)\rangle - 2\theta \langle y^k - y^{k-1}, \mathbf{A}(x^k - x^*)\rangle$$

$$\leq \left( 1 - \max\left\{ \eta_x \mu_x, \min\left\{ \frac{\eta_x \mu_{yx}^2}{2L_y}, \frac{\mu_{yx}^2}{2L_{xy}^2} \right\} \right\} \right) \frac{1}{\eta_x} \|x^k - x^*\|^2$$

$$+ \left( 1 - \max\left\{ \eta_y \mu_y, \min\left\{ \frac{\eta_y \mu_{xy}^2}{2L_x}, \frac{\mu_{xy}^2}{2L_{xy}^2} \right\} \right\} \right) \frac{1}{\eta_y} \|y^k - y^*\|^2$$

$$+ \frac{\theta}{4\eta_y} \|y^k - y^{k-1}\|^2 - \frac{1}{4\eta_y} \|y^{k+1} - y^k\|^2 + \left( \frac{2}{\sigma_x} - 1 \right) \mathrm{D}_f(x_f^k, x^*) + \left( \frac{2}{\sigma_y} - 1 \right) \mathrm{D}_g(y_f^k, y^*)$$

$$+ 2\langle y^{k+1} - y^k, \mathbf{A}(x^{k+1} - x^*)\rangle - 2\theta \langle y^k - y^{k-1}, \mathbf{A}(x^k - x^*)\rangle.$$

Using the definition of $\theta$ we get

$$(\text{LHS}) \leq \theta \left( \frac{1}{\eta_x} \|x^k - x^*\|^2 + \frac{1}{\eta_y} \|y^k - y^*\|^2 + \frac{1}{4\eta_y} \|y^k - y^{k-1}\|^2 - 2\langle y^k - y^{k-1}, \mathbf{A}(x^k - x^*)\rangle \right)$$

$$+ \theta \left( \frac{2}{\sigma_x} \mathrm{D}_f(x_f^k, x^*) + \frac{2}{\sigma_y} \mathrm{D}_g(y_f^k, y^*) \right) - \frac{1}{4\eta_y} \|y^{k+1} - y^k\|^2 + 2\langle y^{k+1} - y^k, \mathbf{A}(x^{k+1} - x^*)\rangle.$$

After rearranging and using the definition of $\Psi^k$ we get

$$\Psi^{k+1} \leq \theta \Psi^k.$$

Finally, using the definition of $\Psi^k$, $\eta_x$ and $\eta_y$ we get

$$
\begin{aligned}
\Psi^k &\geq \frac{1}{\eta_x}\|x^k - x^*\|^2 + \frac{1}{\eta_y}\|y^k - y^*\|^2 + \frac{1}{4\eta_y}\|y^k - y^{k-1}\|^2 - 2\langle y^k - y^{k-1}, \mathbf{A}(x^k - x^*)\rangle \\
&\geq \frac{1}{\eta_x}\|x^k - x^*\|^2 + \frac{1}{\eta_y}\|y^k - y^*\|^2 + \frac{1}{4\eta_y}\|y^k - y^{k-1}\|^2 - 2L_{xy}\|y^k - y^{k-1}\|\|x^k - x^*\| \\
&\geq \frac{1}{\eta_x}\|x^k - x^*\|^2 + \frac{1}{\eta_y}\|y^k - y^*\|^2 + \frac{1}{4\eta_y}\|y^k - y^{k-1}\|^2 - \frac{1}{2\sqrt{\eta_x\eta_y}}\|y^k - y^{k-1}\|\|x^k - x^*\| \\
&\geq \frac{1}{\eta_x}\|x^k - x^*\|^2 + \frac{1}{\eta_y}\|y^k - y^*\|^2 + \frac{1}{4\eta_y}\|y^k - y^{k-1}\|^2 - \frac{1}{4\eta_x}\|x^k - x^*\|^2 - \frac{1}{4\eta_y}\|y^k - y^{k-1}\|^2 \\
&= \frac{3}{4\eta_x}\|x^k - x^*\|^2 + \frac{1}{\eta_y}\|y^k - y^*\|^2.
\end{aligned}
$$

$\square$

*Proof of Theorem 1.* From (64) and (65) we can conclude that

$$
\frac{3}{4\eta_x}\|x^k - x^*\|^2 + \frac{1}{\eta_y}\|y^k - y^*\|^2 \leq \theta^k \Psi^0.
$$

This implies the following inequality

$$
\max\left\{\|x^k - x^*\|^2, \|y^k - x^*\|^2\right\} \leq \theta^k \Psi^0 \max\left\{4\eta_x/3, \eta_y\right\}.
$$

Hence, we can conclude that

$$
\max\left\{\|x^k - x^*\|^2, \|y^k - x^*\|^2\right\} \leq \epsilon,
$$

as long as the number of iterations $k$ satisfies

$$
k \geq \frac{1}{1-\theta}\log\frac{C}{\epsilon},
$$

where $C = \Psi^0 \max\{4\eta_x/3, \eta_y\}$, which does not depend on $\epsilon$. From (58) we obtain

$$
\frac{1}{1-\theta} = \min\left\{\frac{1}{\rho_a(\delta, \sigma_x, \sigma_y)}, \frac{1}{\rho_b(\delta, \sigma_x, \sigma_y)}, \frac{1}{\rho_c(\delta, \sigma_x, \sigma_y)}, \frac{1}{\rho_d(\delta, \sigma_x, \sigma_y)}\right\}.
$$

We can now try to approximately optimize parameters $\delta > 0$ and $\sigma_x, \sigma_y \in (0, 1]$ to obtain the smallest possible values of $\rho_a(\delta, \sigma_x, \sigma_y)^{-1}, \rho_b(\delta, \sigma_x, \sigma_y)^{-1}, \rho_c(\delta, \sigma_x, \sigma_y)^{-1}, \rho_d(\delta, \sigma_x, \sigma_y)^{-1}$. This can be done in a closed form and the result is the following:

$$
\frac{1}{\rho_a} \leq 4 + 4\max\left\{\sqrt{\frac{L_x}{\mu_x}}, \sqrt{\frac{L_y}{\mu_y}}, \frac{L_{xy}}{\sqrt{\mu_x\mu_y}}\right\} \text{ for } \delta = \sqrt{\frac{\mu_y}{\mu_x}}, \sigma_x = \sqrt{\frac{\mu_x}{2L_x}}, \sigma_y = \sqrt{\frac{\mu_x}{2L_x}},
$$

$$
\frac{1}{\rho_b} \leq 4 + 8\max\left\{\frac{\sqrt{L_xL_y}}{\mu_{xy}}, \frac{L_{xy}}{\mu_{xy}}\sqrt{\frac{L_x}{\mu_x}}, \frac{L_{xy}^2}{\mu_{xy}^2}\right\} \text{ for } \delta = \sqrt{\frac{\mu_{xy}^2}{2\mu_xL_x}}, \sigma_x = \sqrt{\frac{\mu_x}{2L_x}}, \sigma_y = \min\left\{1, \sqrt{\frac{\mu_{xy}^2}{4L_xL_y}}\right\},
$$

$$
\frac{1}{\rho_c} \leq 4 + 8\max\left\{\frac{\sqrt{L_xL_y}}{\mu_{yx}}, \frac{L_{xy}}{\mu_{yx}}\sqrt{\frac{L_y}{\mu_y}}, \frac{L_{xy}^2}{\mu_{yx}^2}\right\} \text{ for } \delta = \sqrt{\frac{2\mu_yL_y}{\mu_{yx}^2}}, \sigma_x = \min\left\{1, \sqrt{\frac{\mu_{yx}^2}{4L_xL_y}}\right\}, \sigma_y = \sqrt{\frac{\mu_y}{2L_y}},
$$

$$
\frac{1}{\rho_d} \leq 2 + 8\max\left\{\frac{\sqrt{L_xL_y}L_{xy}}{\mu_{xy}\mu_{yx}}, \frac{L_{xy}^2}{\mu_{yx}^2}, \frac{L_{xy}^2}{\mu_{xy}^2}\right\} \text{ for } \delta = \frac{\mu_{xy}}{\mu_{yx}}\sqrt{\frac{L_y}{L_x}}, \sigma_x = \min\left\{1, \sqrt{\frac{\mu_{yx}^2}{4L_xL_y}}\right\}, \sigma_y = \min\left\{1, \sqrt{\frac{\mu_{xy}^2}{4L_xL_y}}\right\}.
$$

Note, that we set $\mu_y = 0$ in the bound for $\rho_b^{-1}$, $\mu_x = 0$ in the bound for $\rho_c^{-1}$ and $\mu_x = \mu_y = 0$ in the bound for $\rho_d^{-1}$. This is a valid move, because any convex function is 0-strongly convex by the definition of strong convexity. $\square$

# D Proof of Theorem 2

**Lemma 6.** *Problem* (11) *has a unique solution* $(x^*, y^*)$.

*Proof.* Consider operator $T \colon \mathbb{R}^{d_x} \times \mathbb{R}^{d_y} \to \mathbb{R}^{d_x} \times \mathbb{R}^{d_y}$ defined as $T \colon (x, y) \mapsto (x - t_x \nabla_x F(x, y), y - t_y \nabla_y F(x, y))$ for some fixed $t_x, t_y > 0$. It is obvious that $(x, y)$ is a fixed point of operator $T$ if and only if $(x, y)$ is a solution to problem (11). If one can show that this operator is contractive, then it has a unique fixed point due to Banach fixed-point theorem. The proof of the fact that $T$ is contractive is similar to the proof of the rest of Theorem 2. $\qquad\square$

**Lemma 7.** *Let* $\eta_x$ *be defined as*

$$\eta_x = \min\left\{ \frac{1}{8L_x}, \frac{\delta}{4L_{xy}} \right\}, \tag{66}$$

*and let* $\eta_y$ *be defined as*

$$\eta_y = \min\left\{ \frac{1}{8L_y}, \frac{1}{4\delta L_{xy}} \right\}, \tag{67}$$

*where* $\delta > 0$ *is a parameter. Let* $\theta$ *be defined as*

$$\theta = \theta(\delta) = 1 - \max\left\{ \rho_a(\delta), \rho_b(\delta), \rho_c(\delta), \rho_d(\delta) \right\}, \tag{68}$$

*where* $\rho_a(\delta), \rho_b(\delta), \rho_c(\delta), \rho_d(\delta)$ *are defined as*

$$\frac{1}{\rho_a(\delta)} = \max\left\{ \frac{8L_x}{\mu_x}, \frac{8L_y}{\mu_y}, \frac{4L_{xy}}{\delta\mu_x}, \frac{4L_{xy}\delta}{\mu_y} \right\}, \tag{69}$$

$$\frac{1}{\rho_b(\delta)} = \max\left\{ \frac{8L_x}{\mu_x}, \frac{512L_xL_y}{\mu_{xy}^2}, \frac{4L_{xy}}{\delta\mu_x}, \frac{256L_xL_{xy}\delta}{\mu_{xy}^2}, \frac{256L_yL_{xy}}{\mu_{xy}^2\delta}, \frac{128L_{xy}^2}{\mu_{xy}^2} \right\}, \tag{70}$$

$$\frac{1}{\rho_c(\delta)} = \max\left\{ \frac{8L_y}{\mu_y}, \frac{512L_xL_y}{\mu_{yx}^2}, \frac{4L_{xy}\delta}{\mu_y}, \frac{256L_xL_{xy}\delta}{\mu_{yx}^2}, \frac{256L_yL_{xy}}{\mu_{yx}^2\delta}, \frac{128L_{xy}^2}{\mu_{yx}^2} \right\}, \tag{71}$$

$$\frac{1}{\rho_d(\delta)} = \max\left\{ \frac{512L_xL_y}{\min\{\mu_{xy}^2, \mu_{yx}^2\}}, \frac{256L_xL_{xy}\delta}{\min\{\mu_{xy}^2, \mu_{yx}^2\}}, \frac{256L_yL_{xy}}{\min\{\mu_{xy}^2, \mu_{yx}^2\}\delta}, \frac{128L_{xy}^2}{\min\{\mu_{xy}^2, \mu_{yx}^2\}} \right\}. \tag{72}$$

*Let* $\Psi^k$ *be the following Lyapunov function:*

$$\Psi^k = \frac{1}{\eta_x}\|x^k - x^*\|^2 + \frac{1}{\eta_y}\|y^k - y^*\|^2 - 2\langle \nabla_x F(x^{k-1}, y^k) - \nabla_x F(x^{k-1}, y^{k-1}), x^k - x^* \rangle + \frac{5}{16\eta_y}\|y^k - y^{k-1}\|^2. \tag{73}$$

*Then, the following inequalities hold*

$$\Psi^k \geq \frac{3}{4\eta_x}\|x^k - x^*\|^2 + \frac{1}{\eta_y}\|y^k - y^*\|^2, \tag{74}$$

$$\Psi^{k+1} \leq \theta\Psi^k. \tag{75}$$

*Proof.* Using Line 5 of the Algorithm 2 we get.

$$\begin{aligned}
\frac{1}{\eta_x}\|x^{k+1} - x^*\|^2 &= \frac{1}{\eta_x}\|x^k - x^*\|^2 + \frac{2}{\eta_x}\langle x^{k+1} - x^k, x^{k+1} - x^* \rangle - \frac{1}{\eta_x}\|x^{k+1} - x^k\|^2 \\
&= \frac{1}{\eta_x}\|x^k - x^*\|^2 - \frac{1}{\eta_x}\|x^{k+1} - x^k\|^2 \\
&\quad - 2\langle \nabla_x F(x^k, y^k) + \theta(\nabla_x F(x^{k-1}, y^k) - \nabla_x F(x^{k-1}, y^{k-1})), x^{k+1} - x^* \rangle \\
&= \frac{1}{\eta_x}\|x^k - x^*\|^2 - \frac{1}{\eta_x}\|x^{k+1} - x^k\|^2 - 2\langle \nabla_x F(x^k, y^{k+1}), x^{k+1} - x^k + x^k - x^* \rangle \\
&\quad + 2\langle \nabla_x F(x^k, y^{k+1}) - \nabla_x F(x^k, y^k), x^{k+1} - x^* \rangle \\
&\quad - 2\theta\langle \nabla_x F(x^{k-1}, y^k) - \nabla_x F(x^{k-1}, y^{k-1}), x^{k+1} - x^* \rangle.
\end{aligned}$$

Using the Assumption 4 we get

$$\frac{1}{\eta_x}\|x^{k+1}-x^*\|^2 \leq \left(\frac{1}{\eta_x}-\mu_x\right)\|x^k-x^*\|^2 + \left(L_x-\frac{1}{\eta_x}\right)\|x^{k+1}-x^k\|^2 - 2(F(x^{k+1},y^{k+1})-F(x^*,y^{k+1}))$$
$$+ 2\langle\nabla_x F(x^k,y^{k+1})-\nabla_x F(x^k,y^k), x^{k+1}-x^*\rangle$$
$$- 2\theta\langle\nabla_x F(x^{k-1},y^k)-\nabla_x F(x^{k-1},y^{k-1}), x^{k+1}-x^*\rangle.$$

Using the Assumption 5 we get

$$\frac{1}{\eta_x}\|x^{k+1}-x^*\|^2 \leq \left(\frac{1}{\eta_x}-\mu_x\right)\|x^k-x^*\|^2 + \left(L_x-\frac{1}{\eta_x}\right)\|x^{k+1}-x^k\|^2 - 2(F(x^{k+1},y^{k+1})-F(x^*,y^{k+1}))$$
$$+ 2\langle\nabla_x F(x^k,y^{k+1})-\nabla_x F(x^k,y^k), x^{k+1}-x^*\rangle - 2\theta\langle\nabla_x F(x^{k-1},y^k)-\nabla_x F(x^{k-1},y^{k-1}), x^k-x^*\rangle$$
$$+ 2L_{xy}\theta\|x^{k+1}-x^k\|\|y^k-y^{k-1}\|.$$

Similarly, we can obtain the following upper-bound on $\frac{1}{\eta_y}\|y^{k+1}-y^*\|^2$:

$$\frac{1}{\eta_y}\|y^{k+1}-y^*\|^2 \leq \left(\frac{1}{\eta_y}-\mu_y\right)\|y^k-y^*\|^2 + \left(L_y-\frac{1}{\eta_y}\right)\|y^{k+1}-y^k\|^2 + 2(F(x^{k+1},y^{k+1})-F(x^{k+1},y^*)).$$

Summing up the upper-bounds on $\frac{1}{\eta_x}\|x^{k+1}-x^*\|^2$ and $\frac{1}{\eta_y}\|y^{k+1}-y^*\|^2$ gives

$$\text{(LHS)} \leq \left(\frac{1}{\eta_x}-\mu_x\right)\|x^k-x^*\|^2 + \left(L_x-\frac{1}{\eta_x}\right)\|x^{k+1}-x^k\|^2$$
$$+ \left(\frac{1}{\eta_y}-\mu_y\right)\|y^k-y^*\|^2 + \left(L_y-\frac{1}{\eta_y}\right)\|y^{k+1}-y^k\|^2$$
$$+ 2L_{xy}\theta\|x^{k+1}-x^k\|\|y^k-y^{k-1}\| - 2\theta\langle\nabla_x F(x^{k-1},y^k)-\nabla_x F(x^{k-1},y^{k-1}), x^k-x^*\rangle$$
$$+ 2(F(x^*,y^{k+1})-F(x^{k+1},y^*)),$$

where (LHS) is defined as

$$\text{(LHS)} = \frac{1}{\eta_x}\|x^{k+1}-x^*\|^2 + \frac{1}{\eta_y}\|y^{k+1}-y^*\|^2 - 2\langle\nabla_x F(x^k,y^{k+1})-\nabla_x F(x^k,y^k), x^{k+1}-x^*\rangle.$$

The Assumption 4 states, that function $F(x,y)$ is $L_x$-smooth in $x$ and $L_y$-smooth in $y$. Hence, using the optimality conditions (40) we get

$$\text{(LHS)} \leq \left(\frac{1}{\eta_x}-\mu_x\right)\|x^k-x^*\|^2 + \left(L_x-\frac{1}{\eta_x}\right)\|x^{k+1}-x^k\|^2$$
$$+ \left(\frac{1}{\eta_y}-\mu_y\right)\|y^k-y^*\|^2 + \left(L_y-\frac{1}{\eta_y}\right)\|y^{k+1}-y^k\|^2$$
$$+ 2L_{xy}\theta\|x^{k+1}-x^k\|\|y^k-y^{k-1}\| - 2\theta\langle\nabla_x F(x^{k-1},y^k)-\nabla_x F(x^{k-1},y^{k-1}), x^k-x^*\rangle$$
$$- 2(F(x^{k+1},y^*)-F(x^*,y^*)) - 2(F(x^*,y^*)-F(x^*,y^{k+1}))$$
$$\leq \left(\frac{1}{\eta_x}-\mu_x\right)\|x^k-x^*\|^2 + \left(L_x-\frac{1}{\eta_x}\right)\|x^{k+1}-x^k\|^2$$
$$+ \left(\frac{1}{\eta_y}-\mu_y\right)\|y^k-y^*\|^2 + \left(L_y-\frac{1}{\eta_y}\right)\|y^{k+1}-y^k\|^2$$
$$+ 2L_{xy}\theta\|x^{k+1}-x^k\|\|y^k-y^{k-1}\| - 2\theta\langle\nabla_x F(x^{k-1},y^k)-\nabla_x F(x^{k-1},y^{k-1}), x^k-x^*\rangle$$
$$- \frac{\delta_x}{L_x}\|\nabla_x F(x^{k+1},y^*)\|^2 - \frac{\delta_y}{L_y}\|\nabla_y F(x^*,y^{k+1})\|^2,$$

where $\delta_x, \delta_y \in (0, 1]$ are some parameters, that will be defined later. Using the Assumption 6 we get

$$(\text{LHS}) \leq \left(\frac{1}{\eta_x} - \mu_x\right)\|x^k - x^*\|^2 + \left(L_x - \frac{1}{\eta_x}\right)\|x^{k+1} - x^k\|^2$$

$$+ \left(\frac{1}{\eta_y} - \mu_y\right)\|y^k - y^*\|^2 + \left(L_y - \frac{1}{\eta_y}\right)\|y^{k+1} - y^k\|^2$$

$$+ 2L_{xy}\theta\|x^{k+1} - x^k\|\|y^k - y^{k-1}\| - 2\theta\langle\nabla_x F(x^{k-1}, y^k) - \nabla_x F(x^{k-1}, y^{k-1}), x^k - x^*\rangle$$

$$- \frac{\delta_x}{2L_x}\|\nabla_x F(x^{k+1}, y^*) - \nabla_x F(x^{k+1}, y^k)\|^2 + \frac{\delta_x}{L_x}\|\nabla_x F(x^{k+1}, y^k)\|^2$$

$$- \frac{\delta_y}{2L_y}\|\nabla_y F(x^*, y^{k+1}) - \nabla_y F(x^k, y^{k+1})\|^2 + \frac{\delta_y}{L_y}\|\nabla_y F(x^k, y^{k+1})\|^2$$

$$\leq \left(\frac{1}{\eta_x} - \mu_x - \frac{\delta_y \mu_{yx}^2}{2L_y}\right)\|x^k - x^*\|^2 + \left(L_x - \frac{1}{\eta_x}\right)\|x^{k+1} - x^k\|^2$$

$$+ \left(\frac{1}{\eta_y} - \mu_y - \frac{\delta_x \mu_{xy}^2}{2L_x}\right)\|y^k - y^*\|^2 + \left(L_y - \frac{1}{\eta_y}\right)\|y^{k+1} - y^k\|^2$$

$$+ 2L_{xy}\theta\|x^{k+1} - x^k\|\|y^k - y^{k-1}\| - 2\theta\langle\nabla_x F(x^{k-1}, y^k) - \nabla_x F(x^{k-1}, y^{k-1}), x^k - x^*\rangle$$

$$+ \frac{\delta_x}{L_x}\|\nabla_x F(x^{k+1}, y^k)\|^2 + \frac{\delta_y}{L_y}\|\nabla_y F(x^k, y^{k+1})\|^2$$

Using Lines 5 and 6 of the Algorithm 2and the Lipschitzness property of $\nabla_x F(x, y)$ and $\nabla_y F(x, y)$ we get

$$(\text{LHS}) \leq \left(\frac{1}{\eta_x} - \mu_x - \frac{\delta_y \mu_{yx}^2}{2L_y}\right)\|x^k - x^*\|^2 + \left(L_x - \frac{1}{\eta_x}\right)\|x^{k+1} - x^k\|^2$$

$$+ \left(\frac{1}{\eta_y} - \mu_y - \frac{\delta_x \mu_{xy}^2}{2L_x}\right)\|y^k - y^*\|^2 + \left(L_y - \frac{1}{\eta_y}\right)\|y^{k+1} - y^k\|^2$$

$$+ 2L_{xy}\theta\|x^{k+1} - x^k\|\|y^k - y^{k-1}\| - 2\theta\langle\nabla_x F(x^{k-1}, y^k) - \nabla_x F(x^{k-1}, y^{k-1}), x^k - x^*\rangle$$

$$+ \frac{2\delta_x}{L_x}\|\nabla_x F(x^{k+1}, y^k) - \nabla_x F(x^k, y^k) - \theta(\nabla_x F(x^{k-1}, y^k) - \nabla_x F(x^{k-1}, y^{k-1}))\|^2 + \frac{2\delta_x}{L_x\eta_x^2}\|x^{k+1} - x^k\|^2$$

$$+ \frac{2\delta_y}{L_y}\|\nabla_y F(x^k, y^{k+1}) - \nabla_y F(x^{k+1}, y^k)\|^2 + \frac{2\delta_y}{L_y\eta_y^2}\|y^{k+1} - y^k\|^2$$

$$\leq \left(\frac{1}{\eta_x} - \mu_x - \frac{\delta_y \mu_{yx}^2}{2L_y}\right)\|x^k - x^*\|^2 + \left(L_x - \frac{1}{\eta_x}\right)\|x^{k+1} - x^k\|^2$$

$$+ \left(\frac{1}{\eta_y} - \mu_y - \frac{\delta_x \mu_{xy}^2}{2L_x}\right)\|y^k - y^*\|^2 + \left(L_y - \frac{1}{\eta_y}\right)\|y^{k+1} - y^k\|^2$$

$$+ 2L_{xy}\theta\|x^{k+1} - x^k\|\|y^k - y^{k-1}\| - 2\theta\langle\nabla_x F(x^{k-1}, y^k) - \nabla_x F(x^{k-1}, y^{k-1}), x^k - x^*\rangle$$

$$+ 4\delta_x L_x\|x^{k+1} - x^k\|^2 + \frac{4\delta_x L_{xy}^2 \theta^2}{L_x}\|y^k - y^{k-1}\|^2 + \frac{2\delta_x}{L_x\eta_x^2}\|x^{k+1} - x^k\|^2$$

$$+ 4\delta_y L_y\|y^{k+1} - y^k\|^2 + \frac{4\delta_y L_{xy}^2}{L_y}\|x^{k+1} - x^k\|^2 + \frac{2\delta_y}{L_y\eta_y^2}\|y^{k+1} - y^k\|^2.$$

Now, we set $\delta_x = \min\{1, c_x \eta_x L_x\}$, $\delta_y = \min\{1, c_y \eta_y L_y\}$, where $c_x, c_y > 0$ will be defined later, and obtain

$$(\text{LHS}) \leq \left(\frac{1}{\eta_x} - \mu_x - \frac{\delta_y \mu_{yx}^2}{2L_y}\right)\|x^k - x^*\|^2 + \left(L_x - \frac{1}{\eta_x}\right)\|x^{k+1} - x^k\|^2$$

$$+ \left(\frac{1}{\eta_y} - \mu_y - \frac{\delta_x \mu_{xy}^2}{2L_x}\right)\|y^k - y^*\|^2 + \left(L_y - \frac{1}{\eta_y}\right)\|y^{k+1} - y^k\|^2$$

$$+ 2L_{xy}\theta\|x^{k+1} - x^k\|\|y^k - y^{k-1}\| - 2\theta\langle\nabla_x F(x^{k-1}, y^k) - \nabla_x F(x^{k-1}, y^{k-1}), x^k - x^*\rangle$$

$$+ 4c_x\eta_x L_x^2\|x^{k+1} - x^k\|^2 + 4c_x\eta_x L_{xy}^2\theta^2\|y^k - y^{k-1}\|^2 + \frac{2c_x}{\eta_x}\|x^{k+1} - x^k\|^2$$

$$+ 4c_y\eta_y L_y^2\|y^{k+1} - y^k\|^2 + 4c_y\eta_y L_{xy}^2\|x^{k+1} - x^k\|^2 + \frac{2c_y}{\eta_y}\|y^{k+1} - y^k\|^2.$$

Using the definition of $\eta_x$ and $\eta_y$ we get

$$\text{(LHS)} \leq \left(\frac{1}{\eta_x} - \mu_x - \frac{\delta_y\mu_{yx}^2}{2L_y}\right)\|x^k - x^*\|^2 + \left(L_x - \frac{1}{\eta_x}\right)\|x^{k+1} - x^k\|^2$$

$$+ \left(\frac{1}{\eta_y} - \mu_y - \frac{\delta_x\mu_{xy}^2}{2L_x}\right)\|y^k - y^*\|^2 + \left(L_y - \frac{1}{\eta_y}\right)\|y^{k+1} - y^k\|^2$$

$$- 2\theta\langle\nabla_x F(x^{k-1}, y^k) - \nabla_x F(x^{k-1}, y^{k-1}), x^k - x^*\rangle$$

$$+ 4c_x\eta_x L_x^2\|x^{k+1} - x^k\|^2 + \frac{(c_x + 1)\theta^2}{4\eta_y}\|y^k - y^{k-1}\|^2 + \frac{2c_x}{\eta_x}\|x^{k+1} - x^k\|^2$$

$$+ 4c_y\eta_y L_y^2\|y^{k+1} - y^k\|^2 + \frac{c_y + 1}{4\eta_x}\|x^{k+1} - x^k\|^2 + \frac{2c_y}{\eta_y}\|y^{k+1} - y^k\|^2.$$

Now, we choose $c_x = c_y = \frac{1}{4}$ and get

$$\text{(LHS)} \leq \left(\frac{1}{\eta_x} - \mu_x - \frac{\delta_y\mu_{yx}^2}{2L_y}\right)\|x^k - x^*\|^2 + \left(L_x - \frac{1}{\eta_x}\right)\|x^{k+1} - x^k\|^2$$

$$+ \left(\frac{1}{\eta_y} - \mu_y - \frac{\delta_x\mu_{xy}^2}{2L_x}\right)\|y^k - y^*\|^2 + \left(L_y - \frac{1}{\eta_y}\right)\|y^{k+1} - y^k\|^2$$

$$- 2\theta\langle\nabla_x F(x^{k-1}, y^k) - \nabla_x F(x^{k-1}, y^{k-1}), x^k - x^*\rangle$$

$$+ \eta_x L_x^2\|x^{k+1} - x^k\|^2 + \frac{5\theta^2}{16\eta_y}\|y^k - y^{k-1}\|^2 + \frac{1}{2\eta_x}\|x^{k+1} - x^k\|^2$$

$$+ \eta_y L_y^2\|y^{k+1} - y^k\|^2 + \frac{5}{16\eta_x}\|x^{k+1} - x^k\|^2 + \frac{1}{2\eta_y}\|y^{k+1} - y^k\|^2$$

$$= \left(\frac{1}{\eta_x} - \mu_x - \frac{\delta_y\mu_{yx}^2}{2L_y}\right)\|x^k - x^*\|^2 + \frac{\eta_x L_x + \eta_x^2 L_x^2 - 3/16}{\eta_x}\|x^{k+1} - x^k\|^2$$

$$+ \left(\frac{1}{\eta_y} - \mu_y - \frac{\delta_x\mu_{xy}^2}{2L_x}\right)\|y^k - y^*\|^2 + \frac{\eta_y L_y + \eta_y^2 L_y^2 - 3/16}{\eta_y}\|y^{k+1} - y^k\|^2$$

$$- 2\theta\langle\nabla_x F(x^{k-1}, y^k) - \nabla_x F(x^{k-1}, y^{k-1}), x^k - x^*\rangle + \frac{5\theta^2}{16\eta_y}\|y^k - y^{k-1}\|^2 - \frac{5}{16\eta_y}\|y^{k+1} - y^k\|^2.$$

Using the definition of $\eta_x$ and $\eta_y$ we get

$$\text{(LHS)} \leq \left(\frac{1}{\eta_x} - \mu_x - \frac{\delta_y\mu_{yx}^2}{2L_y}\right)\|x^k - x^*\|^2 + \left(\frac{1}{\eta_y} - \mu_y - \frac{\delta_x\mu_{xy}^2}{2L_x}\right)\|y^k - y^*\|^2$$

$$- 2\theta\langle\nabla_x F(x^{k-1}, y^k) - \nabla_x F(x^{k-1}, y^{k-1}), x^k - x^*\rangle + \frac{5\theta^2}{16\eta_y}\|y^k - y^{k-1}\|^2 - \frac{5}{16\eta_y}\|y^{k+1} - y^k\|^2.$$

Using the definition of $\delta_x$ and $\delta_y$ we get

$$\text{(LHS)} \leq \left(1 - \max\left\{\eta_x\mu_x, \min\left\{\frac{\eta_x\mu_{yx}^2}{2L_y}, \frac{\eta_x\eta_y\mu_{yx}^2}{8}\right\}\right\}\right)\frac{1}{\eta_x}\|x^k - x^*\|^2$$

$$+ \left(1 - \max\left\{\eta_y\mu_y, \min\left\{\frac{\eta_y\mu_{xy}^2}{2L_x}, \frac{\eta_y\eta_x\mu_{xy}^2}{8}\right\}\right\}\right)\frac{1}{\eta_y}\|y^k - y^*\|^2$$

$$- 2\theta\langle\nabla_x F(x^{k-1}, y^k) - \nabla_x F(x^{k-1}, y^{k-1}), x^k - x^*\rangle + \frac{5\theta^2}{16\eta_y}\|y^k - y^{k-1}\|^2 - \frac{5}{16\eta_y}\|y^{k+1} - y^k\|^2.$$

Using the definition of $\eta_x, \eta_y$ and $\theta$ we get

$$(\text{LHS}) \le \frac{\theta}{\eta_x} \|x^k - x^*\|^2 + \frac{\theta}{\eta_y} \|y^k - y^*\|^2 - 2\theta \langle \nabla_x F(x^{k-1}, y^k) - \nabla_x F(x^{k-1}, y^{k-1}), x^k - x^* \rangle + \frac{5\theta}{16\eta_y} \|y^k - y^{k-1}\|^2$$
$$- \frac{5}{16\eta_y} \|y^{k+1} - y^k\|^2.$$

After rearranging and using the definition of $\Psi^k$ we get

$$\Psi^{k+1} \le \theta \Psi^k.$$

Finally, using the definition of $\Psi^k$, $\eta_x$ and $\eta_y$ we get

$$\Psi^k = \frac{1}{\eta_x} \|x^k - x^*\|^2 + \frac{1}{\eta_y} \|y^k - y^*\|^2 - 2\langle \nabla_x F(x^{k-1}, y^k) - \nabla_x F(x^{k-1}, y^{k-1}), x^k - x^* \rangle + \frac{5}{16\eta_y} \|y^k - y^{k-1}\|^2$$
$$\ge \frac{1}{\eta_x} \|x^k - x^*\|^2 + \frac{1}{\eta_y} \|y^k - y^*\|^2 - 2L_{xy} \|y^k - y^{k-1}\| \|x^k - x^*\| + \frac{5}{16\eta_y} \|y^k - y^{k-1}\|^2$$
$$\ge \frac{1}{\eta_x} \|x^k - x^*\|^2 + \frac{1}{\eta_y} \|y^k - y^*\|^2 - \frac{1}{4\eta_x} \|x^k - x^*\|^2 - \frac{1}{4\eta_y} \|y^k - y^{k-1}\|^2 + \frac{5}{16\eta_y} \|y^k - y^{k-1}\|^2$$
$$\ge \frac{3}{4\eta_x} \|x^k - x^*\|^2 + \frac{1}{\eta_y} \|y^k - y^*\|^2.$$

$\square$

*Proof of Theorem 2.* From (74) and (75) we can conclude that

$$\frac{3}{4\eta_x} \|x^k - x^*\|^2 + \frac{1}{\eta_y} \|y^k - y^*\|^2 \le \theta^k \Psi^0.$$

This implies the following inequality

$$\max \left\{ \|x^k - x^*\|^2, \|y^k - x^*\|^2 \right\} \le \theta^k \Psi^0 \max \left\{ 4\eta_x/3, \eta_y \right\}.$$

Hence, we can conclude that

$$\max \left\{ \|x^k - x^*\|^2, \|y^k - x^*\|^2 \right\} \le \epsilon,$$

as long as the number of iterations $k$ satisfies

$$k \ge \frac{1}{1 - \theta} \log \frac{C}{\epsilon},$$

where $C = \Psi^0 \max \left\{ 4\eta_x/3, \eta_y \right\}$, which does not depend on $\epsilon$. From (68) we obtain

$$\frac{1}{1 - \theta} = \min \left\{ \frac{1}{\rho_a(\delta)}, \frac{1}{\rho_b(\delta)}, \frac{1}{\rho_c(\delta)}, \frac{1}{\rho_d(\delta)} \right\}.$$

Now, we find the parameter $\delta$ to obtain the following upper bounds on $\rho_a(\delta), \rho_b(\delta), \rho_c(\delta), \rho_d(\delta)$:

$$\frac{1}{\rho_a} = \max \left\{ \frac{8L_x}{\mu_x}, \frac{8L_y}{\mu_y}, \frac{4L_{xy}}{\sqrt{\mu_x \mu_y}} \right\} \text{ for } \delta = \sqrt{\frac{\mu_y}{\mu_x}}, \tag{76}$$

$$\frac{1}{\rho_b} = \max \left\{ \frac{8L_x}{\mu_x}, \frac{512 L_x L_y}{\mu_{xy}^2}, \frac{128 L_{xy}^2}{\mu_{xy}^2} \right\} \text{ for } \delta = \max \left\{ \frac{\mu_{xy}}{8\sqrt{\mu_x L_x}}, \sqrt{\frac{L_y}{L_x}} \right\}, \tag{77}$$

$$\frac{1}{\rho_c} = \max \left\{ \frac{8L_y}{\mu_y}, \frac{512 L_x L_y}{\mu_{yx}^2}, \frac{128 L_{xy}^2}{\mu_{yx}^2} \right\} \text{ for } \delta = \min \left\{ \frac{8\sqrt{\mu_y L_y}}{\mu_{yx}}, \sqrt{\frac{L_y}{L_x}} \right\}, \tag{78}$$

$$\frac{1}{\rho_d} = \max \left\{ \frac{512 L_x L_y}{\mu_{xy}^2}, \frac{512 L_x L_y}{\mu_{yx}^2}, \frac{128 L_{xy}^2}{\mu_{xy}^2}, \frac{128 L_{xy}^2}{\mu_{yx}^2} \right\} \text{ for } \delta = \sqrt{\frac{L_y}{L_x}}. \tag{79}$$

$\square$