# OpenReview forum: "Accelerated Primal-Dual Gradient Method for Smooth and Convex-Concave Saddle-Point Problems with Bilinear Coupling"
_NeurIPS.cc/2022/Conference — NeurIPS 2022 Accept_

### Official Review · Reviewer_DLDA · 2022-07-08

**Rating:** 7
**Confidence:** 3
**Soundness:** 3 good
**Presentation:** 2 fair
**Contribution:** 3 good

**Summary:**

This paper presents an accelerated primal-dual algorithm for smooth convex-concave saddle-point problems of the form
$$
\min_{x}\max_{y}F(x,y)=f(x)+y^{\top}Ax-g(y).
$$
The proposed approach exploits the variational inequality formulation of this problem with single-valued operator $V(x,y)=[\nabla_{x}F(x,y),-\nabla_{y}F(x,y)]$. The derivation of the algorithm combines a standard Forward-Backward iteration with a linear extrapolation step that essentially allows to perform sequential implementations. The main result of the paper is a linear convergence result, in dependence of problem parameters, which covers the merely convex-concave case.


**Questions:**

Is there any algorithm closely related to the one introduced here?

What is the reason introducing Algorithm 2 when the main algorithm in the paper is (apparently?) completely different?

Is there any reason to restrict to the bilinear saddle point problem? Could the algorithm be formulated for general saddle point problems.

**Limitations:**

no discussion of negative societal impact.

**Strengths And Weaknesses:**

Strengths:

+ The main message of the paper is clearly stated and well related to the existing literature.
+ The authors spent a significant effort to explain the relation of the results of the paper with the existing results.

Weaknesses:

- The derivation of the algorithm is not natural to me. The explanation given in the main paper is not giving any insight in how the iterates are constructed.

- The presentation of the paper is very confusing. The main Algorithm proposed is Algorithm 1. This algorithm is motivated by a combination of Forward-backward steps and a linear extrapolation step. Unfortunately, the derivation of the algorithm is not very transparent, and I am still not very confident that I have understood what is going on here. Appendix A contains a different algorithm (Algorithm 2) whose construction is, however, very clear. I cannot see a direct relationship between the two, so assume the material in the Appendix should be treated as a separate result, independent of the rest of the paper. This is not clear from reading of the paper and leaves the reader puzzled how one should read Appendix A. Even worse, Algorithm2 is an alternating versions of the operator extrapolation technique that has been studied recently in a sequence of papers. One important contribution is Simple and optimal methods for stochastic variational inequalities, I: operator extrapolation, https://arxiv.org/abs/2011.02987. A Bregman extension can be found in Hui Zhang,
Extragradient and extrapolation methods with generalized Bregman distances for saddle point problems,
Operations Research Letters, Volume 50, Issue 3, 2022, Pages 329-334.

- The proof is very long and technical. Given the short time available for reviews, it is impossible to check for correctness.

---

> ### Author Response · Authors · 2022-08-02
> **Response to Reviewer DLDA (Part 1)**
>
> Thanks for the positive comments about our work.
>
> > The derivation of the algorithm is not natural to me. The explanation given in the main paper is not giving any insight in how the iterates are constructed.
>
> Please can you let us know what details exactly did you not find natural, or which require further explanations in your view? We are of course interested in making the paper as accessible as possible, subject to the paper length constraints, and subject to the constraint that all our main results should still appear in the main body of the paper.
>
>
> Having said that, we do admit that the derivations and explanations are not easily penetrable by anyone. They are aimed at experts and also a bit wider audience interested in saddle point problems. The work is highly technical, which is a reflection of the mathematical difficulty of the problem we solve. Very few people will have the right training to be able to follow it all, let alone find the developments natural - which we believe is natural. We solve several open problems in a technically challenging and highly contested field, and it is not reasonable to expect that a solution to such a problem will not necessarily appear natural.
>
> While we attempted to be as clear as possible, it is really not possible to  make the paper accessible to everyone.
>
>
> > Is there any reason to restrict to the bilinear saddle point problem? Could the algorithm be formulated for general saddle point problems.
>
> It is well known within the optimization community that solving minimax problems with bilinear coupling is much easier than solving general minimax problems. This is due to the fact that functions $f(x)$ and $g(y)$ appear isolated, and are only connected via  the bilinear coupling term, which allows one to separately apply acceleration techniques to functions $f(x)$ and $g(y)$,  and linear extrapolation step to the computation of the backward step for the bilinear coupling part. This can't be done in the non-bilinear case.

---

> ### Author Response · Authors · 2022-08-02
> **Response to Reviewer DLDA (Part 2)**
>
> > What is the reason introducing Algorithm 2 when the main algorithm in the paper is (apparently?) completely different?
> > I cannot see a direct relationship between the two, so assume the material in the Appendix should be treated as a separate result, independent of the rest of the paper. This is not clear from reading of the paper and leaves the reader puzzled how one should read Appendix A.
>
> We are a bit surprised as to why the reviewer is confused about this point. **We maintain the paper makes this clear; we suspect that perhaps the reviewer did not notice some parts of our paper?**
>
> In short, **Algorithm 2 handles the general non-bilinear case, and its development was partially motivated by the development of the more restrictive (in applicability) Algorithm 1, which handles the bilinear case.**
>
> Both algorithms clearly share a similarity of the linear extrapolation step and gradient extrapolation step. Our paper clearly states what similarities both algorithms share (sec A.2, algorithm 2 development). On the other hand, our paper states that Algorithm 2 is an additional contribution. We don't see why it would be a problem to presenting two results that have a lot in common in a single paper.
>
> The key difference is that Algorithm 2 does not use the acceleration technique. Line 425 “In analogy to the development of Algorithm 1”.  We introduce Algorithm 2 to show how some of our ideas can be generalized to the general non-bilinear case. **While Algorithm 1 is the first that can achieve a linear convergence on smooth convex-concave minimax problems with bilinear coupling, Algorithm 2 is the first algorithm that can achieve a linear convergence on general smooth convex-concave minimax problems.** This is stated clearly  in sec 2.2 and sec 2.3, we believe.
>
> If something else is unclear, please do ask! We are of course very interested to make our paper as understandable as possible.
>
> > Is there any algorithm closely related to the one introduced here?
>
> Not that we know of besides the similarities we described in the part where we comment on algorithm development.
>
> > Even worse, Algorithm2 is an alternating versions of the operator extrapolation technique that has been studied recently in a sequence of papers. One important contribution is Simple and optimal methods for stochastic variational inequalities, I: operator extrapolation, https://arxiv.org/abs/2011.02987. A Bregman extension can be found in Hui Zhang, Extragradient and extrapolation methods with generalized Bregman distances for saddle point problems, Operations Research Letters, Volume 50, Issue 3, 2022, Pages 329-334.
>
> All the methods mentioned by the reviewer are generalizations of Optimistic Gradient Descent Ascent (OGDA aka Reflected Forward Backward). However, we did not think of this intuition during the development of the algorithm. In this case one can also view OGDA as an algorithm based on the idea of linear extrapolation step from Chambolle-Pock. We can mention similarity with OGDA in the revised version. **Moreover, even OGDA is not known to have a linear convergence in the convex concave case (to the best of our knowledge), while Alg 2 has. Our analysis technique is also a highly valuable and novel contribution, as expand elsewhere in our response to multiple reviewers.**
>
> > The presentation of the paper is very confusing. The main Algorithm proposed is Algorithm 1. This algorithm is motivated by a combination of Forward-backward steps and a linear extrapolation step. Unfortunately, the derivation of the algorithm is not very transparent, and I am still not very confident that I have understood what is going on here.
>
> **We believe that the paper does  state clearly what techniques are combined. But it is true we did not go as far as providing a step-by-step guide of how to obtain the iterations of the resulting Algorithm 1 based on the techniques which we described. Perhaps this is what the reviewer would want to see; and indeed, this would make the paper even a bit more accessible, especially to audience less familiar with how the acceleration technique works.**

---

> > ### Comment · Reviewer_DLDA · 2022-08-09
> > **reply to the authors**
> >
> > Thanks for your detailed reply to my report. Following your rebuttal I have increased my score to 7.

---

> > > ### Author Response · Authors · 2022-08-09
> > > **Re: reply to the authors**
> > >
> > > Thank you!!!

---

### Official Review · Reviewer_uksx · 2022-07-11

**Rating:** 6
**Confidence:** 4
**Soundness:** 3 good
**Presentation:** 3 good
**Contribution:** 2 fair

**Summary:**

The paper proposes APDG, an accelerated algorithm in solving the smooth convex-concave saddle-point problem. The main contribution is to show linear convergence rate of APDG under various settings: strongly-convex-strongly-concave, strongly-convex-concave with full-column rank matrix A and convex-concave with full rank square matrix A.

**Questions:**

1.	Is it possible to prove the optimality of APDG under strongly-convex-concave and convex-concave setting?
2.	There is a gap between the rate of APDG and the lower bound under bilinear case. Is the rate of APDG tight or the gap is merely an artifact of the analysis techniques?
3.	Compare the rate of APDG with other methods under convex-concave setting.
4.	Provide numerical results to show the empirical performance of APDG.


**Limitations:**

NA.

**Strengths And Weaknesses:**

Pros: APDG combines two techniques, linear extrapolation and Nesterov’s acceleration to obtain linear rate for various saddle-point problems. The rate matches the lower bound under strongly-convex-strongly-concave setting and show improved rate on the setting of strongly-convex-concave with full column rank A.

Cons: The major contribution seems to be the strongly-convex-strongly-concave case, but the improvement is only a small difference of \sqrt{L/L_{xy}} compared to Wang and Li (2020), which seems to be incremental. For strongly-convex-concave setting, it is unclear
whether APDG is optimal.
For convex-concave setting, as far as I could see, this is NOT the first linear convergence result under the smooth convex-concave problem with square full rank matrix A. PDHG, EGM and PPM all can be shown to exhibit linear convergence under this setting. Besides, when A is square and full rank, the results can even be adapted to nonconvex-noncave setting with A is large enough.
APDG is not optimal even in the simple bilinear case. It has a worse dependence on condition number on bilinear case compared with the optimal method.
There is no numerical experiment to demonstrate the empirical benefit of APDG compared with other popular primal-dual methods under these various settings.

---

> ### Author Response · Authors · 2022-08-02
> **Response to Reviewer uksx (Part 1)**
>
> >The major contribution seems to be the strongly-convex-strongly-concave case, but the improvement is only a small difference of \sqrt{L/L_{xy}} compared to Wang and Li (2020), which seems to be incremental.
>
> - Why do you claim that this is small? **This improvement is not bounded form above and can be arbitrarily large!**
>
> - In Table 1 we show the concurrent results of  Wang and Li (2020) by using $\tilde{O}( )$ notation, while describing our result we use simple $O( )$. In other words **our results in this part are at least logarithmically better even when $L = L_{xy}$.** This is significant -- the logarithmic factor alone can easily be an order of magnitude difference, or more.
>
> - This are not the only improvements, however. **Please can we request that you read our detailed *Response to  Reviewer K3so (Part 1)*, where we address the issue of what improvements we obtain?**
>
> - We emphasize that **our approach has led to a completely different way (Catalyst-free!) of constructing the algorithm, which is very significant from several points of view, and is an orthogonal contribution to the actual theoretical complexity results, which are also significant.** On the basis of our algorithm it is possible to develop *stochastic* and *decentralized* versions of the obtained results (see also answers to Reviewer GTc5). It is not known how to obtain these generalizations based on Catalyst-type (proximal envelope type) methods such as the one of Wang and Li (2020) because of the sensitivity of the outer loop to the accuracy of the solution of the inner problem. Stochastic oracle makes it difficult to solve  the inner problem with high accuracy. So, our approach is fundamentally simpler, sharper, and and such allows for new development and extensions!
>
> >For strongly-convex-concave setting, it is unclear whether APDG is optimal.
>
> **We do not know the lower bound in this rather general setup** (in particular cases lower bounds are known, see e.g. Section 5.2), but APDG has the best known complexity bounds in different regimes (Sections 5.2, 5.3). This is  now a very important open problem.
>
> >For convex-concave setting, as far as I could see, this is NOT the first linear convergence result under the smooth convex-concave problem with square full rank matrix $A$. PDHG, EGM and PPM all can be shown to exhibit linear convergence under this setting. Besides, when $A$ is square and full rank, the results can even be adapted to nonconvex-noncave setting with $A$ is large enough. APDG is not optimal even in the simple bilinear case. It has a worse dependence on condition number on bilinear case compared with the optimal method. There is no numerical experiment to demonstrate the empirical benefit of APDG compared with other popular primal-dual methods under these various settings.
>
> - Indeed, for *bilinear* SPP we are not the first and APDG is *not* the best algorithm. We agree with this.
>
> - But we emphasize that **for *non-bilinear* SPP it seems that (to the best of our knowledge) APDG is the first linearly-convergent algorithm** (see also GDAE for general SPP), when $\mu_{xy} > 0 $ (see also answers to Reviewer GTc5)!
>
> - We agree that numerical experiments would be useful in demonstrating this result. We will try to do it by the time of the camera ready deadline.
>
> > Is it possible to prove the optimality of APDG under strongly-convex-concave and convex-concave setting?
>
> - **Yes it is possible!**  We can regularize the problem (e.g. see Devolder O., Glineur F., Nesterov Y. Double smoothing technique for large-scale linearly constrained convex optimization //SIAM Journal on Optimization. – 2012. – V. 22. – No. 2. – P. 702-727). That is to consider that $\varepsilon \lesssim \mu_x$ and $\varepsilon \lesssim \mu_y$. We skip constants here. These constants depend on the unknown distance between the starting point and the solution. But we can estimate it via restarts by using $\sim 8$ times more iterations (e.g. see Gasnikov, A. V., Gasnikova, E. B., Nesterov, Y. E., & Chernov, A. V. (2016). Efficient numerical methods for entropy-linear programming problems. Computational Mathematics and Mathematical Physics, V. 56(4), 514-524). So, the corresponding new bounds can be obtained from the current formulas by replacing $\mu_x = \epsilon/R_x^2$, where $R_x = ||x^0 - x^*||_2$ or/and $\mu_y = \epsilon/R_y^2$, where $R_y = ||y^0 - y^*||_2$.
>
> - If one aims to provide a direct correction of the algorithm (e.g. in step size policy or something like this) without regularization, for the moment we do not have an answer, it is an interesting open question. Thank you for pointing this out.

---

> > ### Comment · Reviewer_uksx · 2022-08-09
> > **Update**
> >
> > Thanks for the response. The response eases some of my concerns and I'm happy to raise my score from 5 to 6. My major concern is still that how meaningful is the improvement over L_{xy}, L_{xx}, L_{yy}. I can see its value, but that seems to be incremental to me. This is the reason that I cannot further raise my score.

---

> ### Author Response · Authors · 2022-08-02
> **Response to Reviewer uksx (Part 2)**
>
> > There is a gap between the rate of APDG and the lower bound under bilinear case. Is the rate of APDG tight or the gap is merely an artifact of the analysis techniques?
>
> **We believe that our rate of APDG in this case is tight given the analysis technique we used.** Note that our rate matches many existing works: Daskalakis et al. (2018); Liang and Stokes (2019); Gidel et al. (2018, 2019); Mishchenko et al. (2020); Mokhtari et al. (2020). To the best of our knowledge, Azizian et al., (2020) is the only work that reaches the lower bound. However, they specifically aim at solving bilinear problems and develop special analysis techniques for this. As we mention in our paper, we do not specifically focus on solving bilinear problems and provide such a result for completeness. It is an interesting open  question to understand whether it is possible to combine our techniques with the techniques of Azizian et al. (2020).
>
> > Compare the rate of APDG with other methods under convex-concave setting.
>
> Please can you tell us what results of you want us  to compare to? We are not aware of results we can compare to. In the general smooth convex-concave case with bilinear coupling, we have result in eq. (36).
>
> > Provide numerical results to show the empirical performance of APDG.
>
> We will do this, but it will take some time, and we are not able to do it by the rebuttal deadline. [ We wish to say though that this comment is not very appropriate in the same way that a comment aimed at an empirical work requesting the addition of some theoretical results would not be appropriate either. Our results are significant as stated, and are of a theoretical nature. ]

---

### Official Review · Reviewer_GTc5 · 2022-07-11

**Rating:** 9
**Confidence:** 4
**Soundness:** 4 excellent
**Presentation:** 4 excellent
**Contribution:** 4 excellent

**Summary:**

This work studies the smooth convex-concave saddle-point problem with a form of separation and bilinear coupling and proposed a new algorithm named Accelerated Primal-Dual Gradient Method. This achieves an optimal linear convergence rate in the strongly-convex-strongly-concave regime, matching the lower complexity bound [Zhang et al., 2021]. It is even more interesting that the proposed algorithm achieves a linear convergence rate when at least one of the functions $f(x)$ and $g(y)$ is nonstrongly convex. Lastly, the authors introduce a linearly convergent algorithm for the general smooth and convex-concave saddle point problem $\min_{x} \max_{y} F(x, y)$ without the requirement of strong convexity or strong concavity.

**Questions:**

The authors seem to have not discussed anything relevant to a stochastic oracle. Is the result extendable to this case based on your algorithm? Some discussions would be helpful, at least.

**Limitations:**

This is a theoretical work and I do not find any limitations.

**Strengths And Weaknesses:**

**Strength**
The results in this paper provide the first accelerated convergence rate that matches the lower bound of [Zhang et al., 2021]. Especially I like the detailed discussion of complexity with literature in Section 5 and algorithm development in Section 6.2 which is educative and can be of independent interest. I enjoyed reading this paper very much.

**Weakness**
I did not have the chance to check all details, but upon my high-level checking the correctness of the main result is promising, and I do not think there is any weakness.

---

> ### Author Response · Authors · 2022-08-02
> **Response to Reviewer GTc5**
>
> Thanks you for the very positive evaluation of our work; we share your enthusiasm and are proud and excited about  these developments. Below we answer your question about the stochastic oracle.
>
> > The authors seem to have not discussed anything relevant to a stochastic oracle. Is the result extendable to this case based on your algorithm? Some discussions would be helpful, at least.
>
> - Indeed, we do not cover this case in this work, which we believe to be more foundational. Only after this work was completed were we able to start thinking about the proper way of handling the stochastic case.
>
> - We have indeed found that the results are extendable to a stochastic oracle and at the same time to the decentralized setup. Note that for all concurrent works it's difficult to do because of Catalyst-loops in the core of the methods. Our approach is loop-free, and this facilitates and significantly simplifies further developments  and enhancements of such a methods in comparison with  other approaches, which are based on the Catalyst trick.
>
> - In particular, it seems that in this way we can improve the results of X. Zhang, N. S. Aybat, and M. Gurbuzbalaban. Robust accelerated primal-dual methods for computing saddle points. arXiv preprint arXiv:2111.12743, 2021 in the deterministic regime.

---

### Official Review · Reviewer_K3so · 2022-07-13

**Rating:** 7
**Confidence:** 2
**Soundness:** 3 good
**Presentation:** 3 good
**Contribution:** 3 good

**Summary:**

- The authors proposed an accelerated primal dual gradient based algorithm that achieves the optimal lower bound, for the convex-concave saddle-point problem.
- The authors proposed an algorithm that converges linearly with relaxed conditions on strong convexity or strong concavity.

**Questions:**

- It will be helpful to provide some potential extensions / future directions to this work.

**Ethics Review Area:**

["I don’t know"]

**Limitations:**

- The proposed algorithms did not actually improve any bounds achieved by existing algorithms.


**Strengths And Weaknesses:**

Strengths:
- Solid theorems and analysis have been provided for the proposed algorithms.
- The proposed algorithm achieves optimal bounds in different scenarios.

Weaknesses:

- The proposed algorithms did not actually improve any bounds achieved by existing algorithms. [UPDATE: This is not a 100% accurate statemnet. I have ignored the log factor in my intial comparison of the bounds. See the authors' comments for details.]
- The proposed algorithm is solely for achieving the lower bound, while falling short of practical usefulness.

---

> ### Author Response · Authors · 2022-08-02
> **Response to Reviewer K3so (Part 1)**
>
> Thank you for the positive evaluation of our paper. Below we address the highlighted weakness, answer the question, and comment on the limitation.
>
> **To the best of our knowledge, APDG is the first algorithm that achieves optimal convergence rate for strongly-convex-strongly-concave minimax problems with different smoothness constants $(L_x,L_y,L_{xy})$ and different strong convexity constants $(\mu_x,\mu_y)$.**
>
> >The proposed algorithms did not actually improve any bounds achieved by existing algorithms.
>
>
>
> - **We can agree with this statement only if one is prepared to ignore our improvements in logarithmic factors, which we believe are significant, however. Perhaps the reviewer missed to notice this?** This misunderstanding possibly happened because of an insufficient visibility of this improvement in our presentation. Indeed, while in Table 1 we indicate the concurrent results of Wang and Li (2020) using $\tilde{O}( )$ notation (the tilde includes logarithmic factors), we describe our result using $O( )$ (which means an absence of any logarithmic factors). The impact of this difference could have easily been overlooked. In other words, our results in this part are at least logarithmically better, and we will stress this more.
>
> - **We also agree that we do not have an improvement in the regime when $L_{x}=L_{y}$ and $\mu_x = \mu_y$. However, in all other regimes our results do improve the existing bounds, and we believe this is important and substantial since the above simplified regime is clearly highly constrained and does not reflect what happens in most applications.** Our results also better because instead of  the $\sqrt{L_{xy} \max ( L_{x},L_{y}, L_{xy} )}$ factor of Wang and Li (2020), we obtain the $L_{xy}$ factor, which is **always better**, and **is much smaller if  $L_{xy} \ll \max (L_{x},L_{y})$.** This is an important improvement on its own. **We also emphasize that this has been achieved using a completely different way (Catalyst-free!) of constructing the algorithm.** On the basis of our algorithmic and analytical innovations, **it is now for the first time possible to develop stochastic and decentralized versions** of the obtained results (see also answer to Reviewer GTc5). It is not known how to obtain these generalizations based on Catalyst-type (proximal envelope type) methods such as those studied by Wang and Li (2020) because of the sensitivity of the outer loop to the accuracy of the solution of the inner problem. Stochastic oracle makes it difficult to solve  the inner problem with a high accuracy.
>
> - Above we've discussed only one regime (small $\mu_{xy}$). If $\mu_{xy}$ is *not* small, we have principally new results for non-bilinear saddle-point problems (SPP). Before this it was only known that we can solve bi-linear SPP in linear time.  See e.g.
>   - G. M. Korpelevich. The extragradient method for finding saddle points and other problems. Matecon, 12:35–49, 1977; Russian original: Economika Mat. Metody, 12(4):747—756, 1976;
>   - Aryan Mokhtari, Asuman Ozdaglar, and Sarath Pattathil. A unified analysis of extra-gradient and optimistic gradient methods for saddle point problems: Proximal point approach. In International Conference on Artificial Intelligence and Statistics, pages 1497–1507. PMLR, 2020;
>   - Ibrahim et al. (2020);
>   - Azizian et al. (2020).
>
> - **For non-bilinear problems without a strong convexity or a strong concavity assumption, as far as we know, there were no results about linear convergence. Our results here are very significant theoretical discovery on their own!** In Section 5.5, we describe the first such result (see also more general results for GDAE algorithm in Appendix). These results have no analogues at all at the moment.
>
>
> > The proposed algorithm is solely for achieving the lower bound, while falling short of practical usefulness.
>
> - We partially but not fully agree with this comment. Indeed, in the current version of the paper we do not discuss numerics. But due to the lack of Catalyst-type loops in the core of our algorithm, we believe that it will definitely work  better in practice than methods  with proximal loops such as those of  Wang and Li (2020), Xie et al. (2021), or Tominin V. et al. On accelerated methods for saddle-point problems with composite structure //arXiv preprint arXiv:2103.09344. – 2021).  Since the best known theoretical concurrent results are mainly based on the Catalyst-type trick (for saddle-point problems this goes back to the work of Lin, T., Jin, C., and Jordan, M. I. (2020)), we expect that our algorithm will work better in practice. Anyway, thank you for pointing this out. We will try to improve the paper in this part in the final version.

---

> > ### Comment · Reviewer_K3so · 2022-08-10
> > **Review updated**
> >
> > Hi, thanks for pointing out the improvement in the bound. I did ignore this in my first attempt of understanding. Review has been updated.

---

> > > ### Author Response · Authors · 2022-08-10
> > > **Re: Review updated**
> > >
> > > Thanks, and thanks for letting us know!

---

> ### Author Response · Authors · 2022-08-02
> **Response to Reviewer K3so (Part 2)**
>
> > It will be helpful to provide some potential extensions / future directions to this work.
>
>
> - Thank you for pointing this out; this is a good idea.
>
> - We plan to add in the final version some remarks about stochastic and decentralized generalizations and about new results related to this topic, which were only obtained this summer (after our submission).
>
> - We also indicate in future work that in contrast to convex optimization, where the oracle call is uniquely associated with the gradient call $\nabla f(x)$, for SPP we have two criteria: numbers of $\nabla_x F(x,y)$-calls and $\nabla_y F(x,y)$-calls (and more variants for SPP with composites that we consider in the paper).
>
> - Re "optimality": In most of the  papers mentioned previously (and, in particular, in all the papers we cited except Tominin V. et al. 2021) a method is optimal according to the worst of the criteria. In Tominin V. et al. On accelerated methods for saddle-point problems with composite structure //arXiv preprint arXiv:2103.09344. – 2021 authors consider criteria separately. But it is still an open problem to develop lower bounds for multi-criterion setup. This is another aspect of future work we will be happy to comment on.

---

### Author Response · Authors · 2022-08-02
**A Message to All Reviewers and the Area Chair**

Dear Reviews,

Thank you for engaging with our paper; we are glad you found the time to read it, and that some you enjoyed reading it! We are excited about our theoretical results, which we believe are very significant.

Here is a very quick summary of the issues raised and the nature of our response (the detailed responses were posted already):

1. **Reviewer K3so (Decision: 7 = Accept):**  This reviewer claimed that our algorithms did not improve any bounds achieved by existing algorithms, which we show is factually incorrect. The reviewer also said that we solely aimed to achieve the lower bound, which is also factually incorrect. Besides this, a comment was made about the practical usefulness of our methods (which is fair, since we did not include experiments) - however, we explain that the structure of our methods makes us feel optimistic about how it would fare against the competing approaches. Finally, we were asked to mention potential extensions, and we have highlighted several exciting new developments enabled by our work; we will include these comments in the paper. In summary, we believe no serious concerns were raised, and if the mentioned issues originally led the reviewer to assign a lower score that would have otherwise been given, we would be delighted if the score could be increased. Thank you!

2. **Reviewer GTc5 (Decision: 9 = Very Strong Accept):** We are very happy with this score; this is exactly how we feel about our discovery! One question was asked about the possibility of an extension to the stochastic case, and we give an answer in the affirmative. In fact, more extensions were enabled by our fundamental new and simpler approach; we comment on these, too. Thank you!

3. **Reviewer uksx (Decision: 5 = Borderline accept):** The reviewer believes our improvement is just a small factor, which we refute. The mentioned factor can be very large, and other improvement factors were not noticed by the reviewer. We explained this. This reviewer then asked about optimality of our methods in some related regimes; in one regime we indeed do not know (outside of the scope of our paper); and in the other regime we believe we know how to show optimality.  The reviewer seems to have missed that for non-bilinear SPP, our method APDG is the first linearly-convergent algorithm, which is a significant result. We were then asked a few clarifying questions (e.g., explaining the gap between the rate of APDG and the lower bound under bilinear case; comparing the rate of APDG with other methods under convex-concave setting). We answered the first, but do not know how to handle the second since no papers with such results were pointed out to us. Finally, we were asked to do some experiments, which we shall do. In summary, we believe this reviewer failed to notice and appreciate the full extent and import of our results, and we hope that our arguments will change his/her mind, and the the other reviewers will be willing to act as champions. We would be delighted if the score would be substantially increased in the light of these considerations. Thanks you!

4. **Reviewer DLDA (Decision: 5 = Borderline accept):** This reviewer was generally confused by our methods and presentation (e.g., "The derivation of the algorithm is not natural to me."). We will try to improve presentation. Besides this, some clarifying questions were asked  (and we answered), some of them very tangential, but we fail to see any criticism of our actual results. We also feely the reviewer failed to appreciate the full breadth and depth of our results - this is apparent from the review when contrasted with the content of our paper. Hence, we do not understand the reasoning behind the low score. We would of course be very glad if the reviewer could reconsider their view of our work. Thank you!

Thanks again to all reviewer and the AC for taking your time to review our work. You are all doing this for free, and this is highly appreciated!

Best regards,

Authors of the submission
(undisclosed location; Aug 2, 2022)

---

### Author Response · Authors · 2022-08-07
**What do you think about our rebuttal?**

Dear reviewers,

Please can you let us know what did you think about our rebuttal? Did we address your concerns and answer your questions to your satisfaction? If not, please can you let us know what points still need to be addressed, and why? We would appreciate if you could let us know, so that we can still respond by the author-feedback deadline, which is now fast approaching.

Thanks you very much for donating your free time to evaluating your paper, especially during the Summer period, when we believe many of you (just like us) are taking some vacation. This is much appreciated and highly valued by the community in general and by us in particular.

Best,

author(s)

---

### Meta-Review · Area_Chair_X2PN · 2022-08-25

**Recommendation:** Accept
**Confidence:** Certain

**Metareview:**

All reviewers liked the paper and the overall impression is very positive - clear accept.

**Award:**

No

---

### Decision · Program_Chairs · 2022-09-14

Accept